# 🙌🙌 OPENHANDS: AN OPEN PLATFORM FOR AI SOFTWARE DEVELOPERS AS GENERALIST AGENTS

Xingyao Wang[1,10], Boxuan Li[2], Yufan Song[2], Frank F. Xu[2], Xiangru Tang[3],
Mingchen Zhuge[6], Jiayi Pan[4], Yueqi Song[2], Bowen Li, Jaskirat Singh[7],
Hoang H. Tran[8], Fuqiang Li, Ren Ma, Mingzhang Zheng, Bill Qian[3], Yanjun Shao[3],
Niklas Muennighoff[5], Yizhe Zhang, Binyuan Hui[9], Junyang Lin[9],
Robert Brennan[10], Hao Peng[1], Heng Ji[1], Graham Neubig[2,10]
[1]UIUC  [2]CMU  [3]Yale  [4]UC Berkeley  [5]Contextual AI  [6]KAUST  [7]ANU
[8]HCMUT  [9]Alibaba  [10]All Hands AI
xingyao6@illinois.edu, gneubig@cs.cmu.edu

## ABSTRACT

Software is one of the most powerful tools that we humans have at our disposal; it allows a skilled programmer to interact with the world in complex and profound ways. At the same time, thanks to improvements in large language models (LLMs), there has also been a rapid development in AI agents that interact with and affect change in their surrounding environments. In this paper, we introduce OpenHands (*f.k.a.* OpenDevin), a platform for the development of powerful and flexible AI agents that interact with the world in similar ways to those of a human developer: by writing code, interacting with a command line, and browsing the web. We describe how the platform allows for the implementation of new agents, safe interaction with sandboxed environments for code execution, coordination between multiple agents, and incorporation of evaluation benchmarks. Based on our currently incorporated benchmarks, we perform an evaluation of agents over 15 challenging tasks, including software engineering (*e.g.*, SWE-BENCH) and web browsing (*e.g.*, WEBARENA), among others. Released under the permissive MIT license, OpenHands is a community project spanning academia and industry with more than 2.1K contributions from over 188 contributors.

|  |  |  |
|---|---|---|
| 🔗 | **Code** | https://github.com/All-Hands-AI/OpenHands |
| 🔗 | **Slack** | http://bit.ly/OpenHands-Slack |

## 1 INTRODUCTION

Powered by large language models (LLMs; OpenAI 2024b; Team et al. 2023; Jiang et al. 2024; Chang et al. 2024), user-facing AI systems (such as ChatGPT) have become increasingly capable of performing complex tasks such as accurately responding to user queries, solving math problems, and generating code. In particular, AI *agents*, systems that can perceive and act upon the external environment, have recently received ever-increasing research focus. They are moving towards performing complex tasks such as developing software (Jimenez et al., 2024), navigating real-world websites (Zhou et al., 2023a), doing household chores (Ahn et al., 2022), or even performing scientific research (Boiko et al., 2023; Tang et al., 2024a).

As AI agents become capable of tackling complex problems, their development and evaluation have also become challenging. There are numerous recent efforts in creating open-source frameworks that facilitate the development of agents (Hong et al., 2023; Chen et al., 2024; Wu et al., 2023). These agent frameworks generally include: 1) **interfaces** through which agents interact with the world (such as JSON-based function calls or code execution), 2) **environments** in which agents operate, and 3) **interaction mechanisms** for human-agent or agent-agent communication. These frameworks streamline and ease the development process in various ways (Tab. 1, §C).

When designing AI agents, we can also consider how *human* interacts with the world. The most powerful way in which humans currently interact with the world is through *software* – software

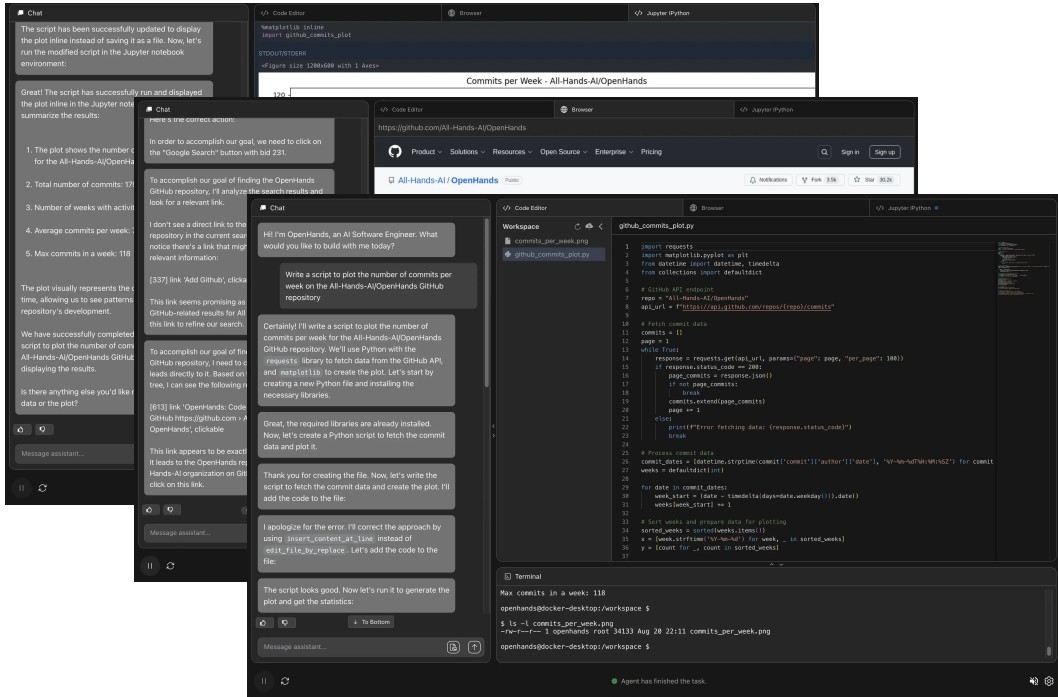

Figure 1: OpenHands User Interface (UI, §D) allows users to view files, check executed bash commands/Python code, observe the agent's browser activity, and directly interact with the agent.

powers every aspect of our life, supporting everything from the logistics for basic needs to the advancement of science, technology, and AI itself. Given the power of software, as well as the existing tooling around its efficient development, use, and deployment, it provides the ideal interface for AI agents to interact with the world in complex ways. However, building agents that can effectively develop software comes with its own unique challenges. How can we enable agents to effectively *create and modify code in complex software systems*? How can we provide them with tools to *gather information on-the-fly* to debug problems or gather task-requisite information? How can we ensure that development is *safe and avoids negative side effects* on the users' systems?

In this paper, we introduce OpenHands (*f.k.a.* OpenDevin), a community-driven platform designed for the development of generalist and specialist AI agents that interact with the world through software.[1] It features:

(1) An **interaction mechanism** which allows user interfaces, agents, and environments to interact through an *event stream* architecture that is powerful and flexible (§2.1).

(2) A **runtime environment** that consists of a docker-sandboxed operating system with a bash shell, a web browser, and IPython server that the agents can interact with (§2.2).

(3) An **interface** allowing the agent to interact with the environment in a manner similar to actual software engineers (§2.3). We provide the capability for agents to a) create and edit complex software, b) execute arbitrary code in the sandbox, and c) browse websites to collect information.

(4) **Multi-agent delegation**, allowing multiple specialized agents to work together (§2.4).

(5) **Evaluation framework**, facilitating the evaluation of agents across a wide range of tasks (§4).

Importantly, OpenHands is not just a conceptual framework, but it also includes a comprehensive and immediately usable implementation of agents, environments, and evaluations. As of this writing, OpenHands includes an agent hub with over 10 implemented agents (§3), including a strong generalist agent implemented based on the CodeAct architecture (Wang et al., 2024a), with additions for web browsing (ServiceNow) and code editing specialists (Yang et al., 2024). Interaction with users is implemented through a chat-based user interface that visualizes the agent's current actions and allows

---

[1]While initially inspired by AI software engineer Devin (Cognition.ai), OpenHands has quickly evolved to support much wider range of applications beyond software engineering through diverse community contributions.

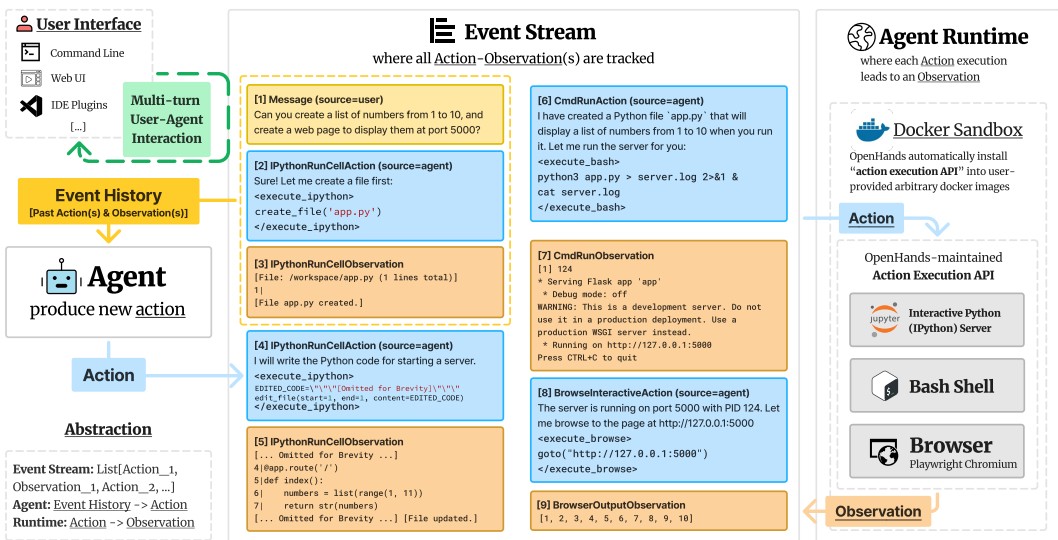

Figure 2: OpenHands consists of 3 main components: 1) **Agent abstraction** where community can contribute different implementation of agents (§2.1) into agenthub (§3); 2) **Event stream** for tracking history of actions and observations; 3) **Runtime** to execute all actions into observations (§2.2).

for real-time feedback (Fig. 1, §D). Furthermore, the evaluation framework currently supports 15 benchmarks, which we use to evaluate our agents (§4).

Released under a permissive MIT license allowing commercial use, OpenHands is poised to support a diverse array of research and real-world applications across academia and industry. OpenHands has gained significant traction, with 32K GitHub stars and more than 2.1K contributions from over 188 contributors. We envision OpenHands as a catalyst for future research innovations and diverse applications driven by a broad community of practitioners.

## 2 OPENHANDS ARCHITECTURE

We next describe using OpenHands in detail. In particular, we discuss 1) how to define and implement an agent (§2.1), 2) how each action execution leads to an observation (§2.2), 3) how to reliably manage and extend commonly used skills for agents (§2.3), and 4) how to compose multiple agents together for task solving (§2.4). Fig. 2 provides an overview.

### 2.1 AGENT DEFINITION AND IMPLEMENTATION

An **agent** can perceive the **state** of the environment (*e.g.*, prior actions and observations) and produce an **action** for execution while solving a user-specified task.

**The State and Event Stream.** In OpenHands, the state is a data structure that encapsulates all relevant information for the agent's execution. A key component of this state is the **event stream**, which is a chronological collection of past actions and observations, including the agent's own actions and user interactions (*e.g.*, instructions, feedback). In addition to the event stream, the state incorporates auxiliary information for agent's operation, such as the accumulative cost of LLM calls, metadata to track multi-agent delegation (§2.4), and other execution-related parameters.

**Actions.** Inspired by CodeAct (Wang et al., 2024a), OpenHands connects an agent with the environment through a core set of general actions. Actions `IPythonRunCellAction` and `CmdRunAction` enable the agent to execute *arbitrary* Python code and bash commands inside the sandbox environment (*e.g.*, a securely isolated Linux operating system). `BrowserInteractiveAction` enables interaction with a web browser with a domain-specific language for browsing introduced by BrowserGym (Drouin et al., 2024). These actions were chosen to provide a comprehensive yet flexible set of primitives covering most tasks performed by human software engineers and analysts. The action space based on programming languages (PL) is powerful

and flexible enough to perform any task with tools in different forms (*e.g.*, Python function, REST API, *etc*.) while being reliable and easy to maintain (Wang et al., 2024a) .

This design is also compatible with existing tool-calling agents that require a list of pre-defined tools (Chase, 2022). That is, users can easily define tools using PL supported in primitive actions (*e.g.*, write a Python function for calculator) and make those tools available to the agent through JSON-style function-calling experiences (Qin et al., 2023). Moreover, the framework's powerful PL-based primitives further make it possible for the agents to create tools by themselves (*e.g.*, by generating Python functions, Yuan et al. 2023) when API to complete the task is unavailable. Refer to §2.3 for how these core PL-based actions can be composed into a diverse set of tools.

Figure 3: Minimal example of implementing an agent in OpenHands.

```python
class MinimalAgent:
    def reset(self) -> None:
        self.system_message = "You are a helpful assistant ..."

    def step(self, state: State):
        messages: list[dict[str, str]] = [
            {'role': 'system', 'content': self.system_message}
        ]
        for prev_action, obs in state.history:
            action_message = get_action_message(prev_action)
            messages.append(action_message)
            obs_message = get_observation_message(obs)
            messages.append(obs_message)

        # use llm to generate response (e.g., thought, action)
        response = self.llm.do_completion(messages)

        # parse and execute action in the runtime
        action = self.parse_response(response)
        if self.is_finish_command(action):
            return AgentFinishAction()
        elif self.is_bash_command(action):
            return CmdRunAction(command=action.command)
        elif self.is_python_code(action):
            return IPythonRunCellAction(code=action.code)
        elif self.is_browser_action(action):
            return BrowseInteractiveAction(code=action.code)
        else:
            return MessageAction(content=action.message)
```

**Observations.** Observations describe the environmental changes (*e.g.*, execution result of prior actions, text messages from the human user *etc*.) that the agent observes.

**Implement a New Agent.** The agent abstraction is designed to be simple yet powerful, allowing users to create and customize agents for various tasks easily. The core of the agent abstraction lies in the `step` function, which takes the current state as input and generates an appropriate action based on the agent's logic. Simplified example code for the agent abstraction is illustrated in Fig. 3. By providing this abstraction, OpenHands allows the users to focus on defining desired agent behavior and logic without worrying about the low-level details of how actions are executed (§2.2).

## 2.2 AGENT RUNTIME: HOW EXECUTION OF ACTIONS RESULTS IN OBSERVATIONS

Agent Runtime provides a general environment that equips the agent with an action space comparable to that of human software developers, enabling OpenHands agents to tackle a wide range of software development and web-based tasks, including complex software development workflows, data analysis projects, web browsing tasks, and more. It allows the agent to access a bash terminal to run code and command line tools, utilize a Jupyter notebook for writing and executing code on-the-fly, and interact with a web browser for web-based tasks (*e.g.*, information seeking).

**Docker Sandbox.** For each task session, OpenHands spins up a securely isolated docker container sandbox, where all the actions from the event stream are executed. OpenHands connects to the sandbox through a REST API server running inside it (i.e., the OpenHands action execution API), executes arbitrary actions (e.g., bash command, python code) from the event stream, and returns the execution results as observations. A configurable workspace directory containing files the user wants the agent to work on is mounted into that secure sandbox for OpenHands agents to access.

**OpenHands Action Execution API.** OpenHands maintains an API server that runs *inside the docker sandbox* to listen for action execution requests from the event stream. The API server maintains:

(1) A bash shell that connects with the operating system environment (specified by the docker image) for command execution.

(2) A Jupyter IPython server to handle interactive *python* (IPython) code execution requests and return the execution results back to the event stream.

(3) A Chromium browser based on Playwright. The provider provides a set of action primitives defined by BrowserGym (ServiceNow; Drouin et al., 2024), such as navigation, clicking, typing, and scrolling. The full set of actions is detailed in §J. After executing these actions, the browser

runtime provides a rich set of observations about the current state of the browser, including HTML, DOM, accessibility tree (Mozilla), screenshot, opened tabs, *etc.*

**Arbitrary Docker Image Support.** OpenHands allows agents to run on arbitrary operating systems with different software environments by supporting runtime based on arbitrary docker images. OpenHands implements a build mechanism that takes a user-provided arbitrary docker image and installs OpenHands action execution API into that image to allow for agent interactions. We include a detailed description of OpenHands agent runtime in §F.

### 2.3 AGENT SKILLS: THE EXTENSIBLE AGENT-COMPUTER INTERFACE

SWE-Agent (Yang et al., 2024) highlights the importance of a carefully crafted Agent-Computer Interface (ACI, *i.e.*, specialized tools for particular tasks) in successfully solving complex tasks. However, creating, maintaining, and distributing a wide array of tools can be a daunting engineering challenge, especially when we want to make these tools available to different agent implementations (§3). To tackle these, we build an **AgentSkills library**, a toolbox designed to enhance the capabilities of agents, offering utilities not readily available through basic *bash* commands or *python* code.

**Easy to create and extend tools.** AgentSkills is designed as a Python package consisting of different utility functions (*i.e.*, tools) that are automatically imported into the Jupyter IPython environment (§2.2). The ease of defining a Python function as a tool lowers the barrier for community members to contribute new tools to the library. The generality of Python packages also allows different agent implementations to easily leverage these tools through one of our core action `IPythonRunCellAction` (§2.1).

**Inclusion criteria and philosophy.** In the AgentSkills library, we do not aim to wrap every possible Python package and re-teach agents their usage (*e.g.*, LLM already knows `pandas` library that can read CSV file, so we don't need to re-create a tool that teaches the agent to read the same file format). We only add a new skill when: (1) it is not readily achievable for LLM to write code directly (*e.g.*, edit code and replace certain lines), and/or (2) it involves calling an external model (*e.g.*, calling a speech-to-text model, or model for code editing (Sanger)).

**Currently supported skills.** AgentSkills library includes file editing utilities adapted from SWE-Agent (Yang et al., 2024) and Aider (Gauthier) like `edit_file`, which allows modifying an existing file from a specified line; scrolling functions `scroll_up` and `scroll_down` for viewing a different part of files. It also contains tools that support reading multi-modal documents, like `parse_image` and `parse_pdf` for extracting information from images using vision-language models (*e.g.*, GPT-4V) and reading text from PDFs, respectively. A complete list of supported skills can be found in §I.

### 2.4 AGENT DELEGATION: COOPERATIVE MULTI-AGENT INTERACTION

OpenHands allows interactions between multiple agents as well. To this end, we use a special action type `AgentDelegateAction`, which enables an agent to delegate a specific subtask to another agent. For example, the generalist CodeActAgent, with limited support for web-browsing, can use `AgentDelegateAction` to delegate web browsing tasks to the specialized BrowsingAgent to perform more complex browsing activity (*e.g.*, navigate the web, click buttons, submit forms, *etc.*).

## 3 AGENTHUB: A HUB OF COMMUNITY-CONTRIBUTED AGENTS

Based on our agent abstraction (§2.1), OpenHands supports a wide range of community-contributed agent implementations for end users to choose from and act as baselines for different agent tasks.

**CodeAct Agent.** CodeActAgent is the default generalist agent based on the CodeAct framework (Wang et al., 2024a). At each step, the agent can (1) converse to communicate with humans in natural language to ask for clarification, confirmation, *etc.*, or (2) to perform the task by executing code (*a.k.a.*, **CodeAct**), including executing bash commands, Python code, or browser-specific programming language (§2.2). This general action space allows the agent (v1.5 and above) to perform various tasks, including editing files, browsing the web, running programs, etc.

Table 1: Comparison of different AI agent frameworks (§C). SWE refers to 'software engineering'. **Standardized tool library**: if framework contains reusable tools for different agent implementations (§2.3); **Built-in sandbox & code execution**: if it supports sandboxed execution of arbitrary agent-generated code; **Built-in web browser**: if it provides agents access to a fully functioning web browser; **Human-AI collaboration**: if it enables multi-turn human-AI collaboration (*e.g.*, human can interrupt the agent during task execution and/or provide additional feedback and instructions); **AgentHub**: if it hosts implementations of various agents (§3); **Evaluation Framework**: if it offers systematic evaluation of implemented agents on challenging benchmarks (§4); **Agent QC** (Quality Control): if the framework integrates tests (§E) to ensure overall framework software quality.

| Framework | Domain | Graphic User Interface | Standardized Tool Library | Built-in Sandbox & Code Execution | Built-in Web Browser | Multi-agent Collaboration | Human-AI Collaboration | AgentHub | Evaluation Framework | Agent QC |
|---|---|---|---|---|---|---|---|---|---|---|
| AutoGPT (Gravitas, 2023) | General | ✔ | ✗ | ✗ | ✗ | ✗ | ✗ | ✔ | ✗ | ✔ |
| LangChain (Chase, 2022) | General | ✗ | ✔ | ✗* | ✗* | ✗ | ✗ | ✔ | ✗ | ✗ |
| MetaGPT (Hong et al., 2023) | General | ✗ | ✔ | ✗ | ✔ | ✔ | ✗ | ✔ | ✗ | ✔ |
| AutoGen (Wu et al., 2023) | General | ✗ | ✔ | ✔ | ✔ | ✔ | ✔ | ✔ | ✔ | ✗ |
| AutoAgents (Chen et al., 2024) | General | ✗ | ✗ | ✗ | ✗ | ✔ | ✗ | ✗ | ✗ | ✗ |
| Agents (Zhou et al., 2023b) | General | ✗ | ✗ | ✗ | ✗ | ✔ | ✔ | ✗ | ✗ | ✗ |
| Xagents (Team, 2023) | General | ✔ | ✔ | ✔ | ✔ | ✔ | ✗ | ✔ | ✗ | ✗ |
| OpenAgents (Xie et al., 2023) | General | ✔ | ✗ | ✔ | ✔ | ✗ | ✔ | ✗ | ✗ | ✗ |
| GPTSwarm (Zhuge et al., 2024) | General | ✗ | ✔ | ✗ | ✗ | ✔ | ✗ | ✗ | ✗ | ✗ |
| AutoCodeRover (Zhang et al., 2024b) | SWE | ✗ | ✗ | ✔ | ✗ | ✗ | ✗ | ✗ | ✗ | ✗ |
| SWE-Agent (Yang et al., 2024) | SWE | ✗ | ✗ | ✔ | ✗ | ✗ | ✗ | ✗ | ✗ | ✗ |
| **OpenHands** | General | ✔ | ✔ | ✔ | ✔ | ✔ | ✔ | ✔ | ✔ | ✔ |

* No native support. Third-party commercial options are available.

**Browsing Agent.** We implemented a generalist web agent called Browsing Agent, to serve as a simple yet effective baseline for web agent tasks. The agent is similar to that in WebArena (Zhou et al., 2023a), but with improved observations and actions, with only zero-shot prompting. Full prompts are in §K.

**GPTSwarm Agent.** GPTSwarm (Zhuge et al., 2024) pioneers the use of optimizable graphs to construct agent systems, unifying language agent frameworks through modularity. Each node represents a distinct operation, while edges define collaboration and communication pathways. This design allows automatic optimization of nodes and edges, driving advancements in creating multi-agent systems.

**Micro Agent(s).** In addition, OpenHands enables the creation of **micro agent**, an agent *specialized* towards a particular task. A micro agent re-uses most implementations from an existing generalist agent (e.g., CodeAct Agent). It is designed to lower the barrier to agent development, where community members can share specialized prompts that work well for their particular use cases.

## 4 EVALUATION

To systematically track progress in building generalist digital agents, as listed in Tab. 2, we integrate 15 established benchmarks into OpenHands. These benchmarks cover software engineering, web browsing, and miscellaneous assistance. In this section, we compare OpenHands to open-source reproducible baselines that do not perform manual prompt engineering specifically based on the benchmark *content*. Please note that we use 'OH' as shorthand for OpenHands for the rest of this section for brevity reasons.

Table 2: Evaluation benchmarks in OpenHands.

| Category | Benchmark | Required Capability |
|---|---|---|
| Software | SWE-Bench (Jimenez et al., 2024) | Fixing Github issues |
| | HumanEvalFix (Muennighoff et al., 2024) | Fixing Bugs |
| | BIRD (Li et al., 2023b) | Text-to-SQL |
| | BioCoder (Tang et al., 2024c) | Bioinformatics coding |
| | ML-Bench (Tang et al., 2024b) | Machine learning coding |
| | Gorilla APIBench (Patil et al., 2023) | Software API calling |
| | ToolQA (Zhuang et al., 2024) | Tool use |
| Web | WebArena (Zhou et al., 2023a) | Goal planning & realistic browsing |
| | MiniWoB++ (Liu et al., 2018) | Short trajectory on synthetic web |
| Misc. Assistance | GAIA (Mialon et al., 2023) | Tool-use, browsing, multi-modality |
| | GPQA (Rein et al., 2023) | Graduate-level Google-proof Q&A |
| | AgentBench (Liu et al., 2023) | Operating system interaction (bash) |
| | MINT (Wang et al., 2024b) | Multi-turn math and code problems |
| | Entity Deduction Arena (Zhang et al., 2024a) | State tracking & strategic planning |
| | ProofWriter (Tafjord et al., 2021) | Deductive Logic Reasoning |

### 4.1 RESULT OVERVIEW

In OpenHands, our goal is to develop **general digital agents** capable of interacting with the world through software interfaces (as exemplified by the code actions described in §2.1). We recognize that a software agent should excel not only in code editing but also in web browsing and various auxiliary tasks, such as answering questions about code repositories or conducting online research.

Table 3: Selected evaluation results for OpenHands agents (§4). See Tab. 4 (software), Tab. 5 (web), Tab. 6 (miscellaneous assistance) for full results across benchmarks.

| Agent | Model | Software (§4.2) SWE-Bench Lite | Web (§4.3) WebArena | Misc. (§4.4) GPQA | GAIA |
|---|---|---|---|---|---|
| *Software Engineering Agents* | | | | | |
| SWE-Agent (Yang et al., 2024) | `gpt-4-1106-preview` | 18.0 | – | – | – |
| AutoCodeRover (Zhang et al., 2024b) | `gpt-4-0125-preview` | 19.0 | – | – | – |
| Aider (Gauthier) | `gpt-4o` & `claude-3-opus` | 26.3 | – | – | – |
| Moatless Tools (Örwall) | `claude-3.5-sonnet` | 26.7 | – | – | – |
| Agentless (Xia et al., 2024) | `gpt-4o` | 27.3 | – | – | – |
| *Web Browsing Agents* | | | | | |
| Lemur (Xu et al., 2023) | `Lemur-chat-70b` | – | 5.3 | – | – |
| Patel et al. (2024) | Trained 72B w/ synthetic data | – | 9.4 | – | – |
| AutoWebGLM (Lai et al., 2024) | Trained 7B w/ human/agent annotation | – | 18.2 | – | – |
| Auto Eval & Refine (Pan et al., 2024) | GPT-4 + Reflexion w/ GPT-4V | – | 20.2 | – | – |
| WebArena Agent (Zhou et al., 2023a) | `gpt-4-turbo` | – | 14.4 | – | – |
| *Misc. Assistance Agents* | | | | | |
| AutoGPT (Gravitas, 2023) | `gpt-4-turbo` | – | – | – | 13.2 |
| Few-shot Prompting + Chain-of-Thought (Rein et al., 2023) | `Llama-2-70b-chat` | – | – | 28.1 | – |
| | `gpt-3.5-turbo-16k` | – | – | 29.6 | – |
| | `gpt-4` | – | – | 38.8 | – |
| **OpenHands Agents** | | | | | |
| CodeActAgent `v1.8` | `gpt-4o-mini-2024-07-18` | 6.3 | 8.3 | – | – |
| | `gpt-4o-2024-05-13` | 22.0 | 14.5 | *53.1 | – |
| | `claude-3-5-sonnet` | 26.0 | 15.3 | 52.0 | – |
| GPTSwarm `v1.0` | `gpt-4o-2024-05-13` | – | – | – | 32.1 |

[*] Numbers are reported from CodeActAgent `v1.5`.

Tab. 3 showcases a curated set of evaluation results. While OpenHands agents may not achieve top performance in every category, they are designed with generality in mind. Notably, the same CodeAct agent, without any modifications to its system prompt, demonstrates competitive performance across three major task categories: software development, web interaction, and miscellaneous tasks. This is particularly significant when compared to the baseline agents, which are typically designed and optimized for specific task categories.

## 4.2 SOFTWARE ENGINEERING

Next, we report results specifically for software engineering benchmarks in Tab. 4.

**SWE-Bench** (Jimenez et al., 2024) is designed to assess agents' abilities in solving real-world GitHub issues, such as bug reports or feature requests. The agent interacts with the repository and attempts to fix the issue provided through file editing and code execution. The agent-modified code repository is tested against a test suite incorporating new tests added from human developers' fixes for the same issue. Each test instance accompanies a piece of "hint text" that consists of natural language suggestions for how to solve the problem. Throughout this paper, we report all results *without using hint text*. A canonical subset, SWE-bench Lite, is created to facilitate accessible and efficient testing. We default to use this subset for testing for cost-saving consideration.[2] **Result.** As shown in Tab. 4, our most recent version of CodeActAgent v1.8, using `claude-3.5-sonnet`, achieves a competitive resolve rate of 26% compared to other open-source SWE specialists.

### 4.2.1 HUMANEVALFIX

**HumanEvalFix** (Muennighoff et al., 2024) tasks agents to fix a bug in a provided function with the help of provided test cases. The bugs are created to ensure one or more test cases fail. We focus on the Python subset of the benchmark and allow models to solve the bugs by self-debug over multiple turns, incorporating feedback from test execution. We follow the setup from Muennighoff et al. (2024) using pass@k (Chen et al., 2021). **Results.** In Tab. 4, OpenHands CodeActAgent successfully fixes 79.3% of bugs in the Python split. This is significantly better than all non-agentic approaches, almost doubling the performance of `StarCoder2-15B` (Lozhkov et al., 2024; Li et al., 2023c). While SWE-Agent achieves 87.7%, Yang et al. (2024) provides the model a full demonstration of a successful sample trajectory fixing one of the bugs in the test dataset ("1-shot"), whereas our evaluation of OpenHands is 0-shot. As HumanEvalFix has been created by humans and all bugs

---

[2]Running the complete set of 2294 instances costs $6.9k, using a conservative estimate of $3 per instance.

Table 4: OpenHands Software Engineering evaluation results (§4.2).

| Agent | Model | Success Rate (%) | $ Avg. Cost |
|---|---|---|---|
| **SWE-Bench Lite** (Jimenez et al., 2024), 300 instances, *w/o Hint* | | | |
| SWE-Agent (Yang et al., 2024) | `gpt-4-1106-preview` | 18.0 | 1.67 |
| AutoCodeRover (Zhang et al., 2024b) | `gpt-4-0125-preview` | 19.0 | – |
| Aider (Gauthier) | `gpt-4o` & `claude-3-opus` | 26.3 | – |
| OH CodeActAgent v1.8 | `gpt-4o-mini-2024-07-18` | 7.0 | 0.01 |
| | `gpt-4o-2024-05-13` | 22.0 | 1.72 |
| | `claude-3-5-sonnet@20240620` | 26.0 | 1.10 |
| **HumanEvalFix** (Muennighoff et al., 2024), 164 instances | | | |
| Prompting, 0-shot | `BLOOMZ-176B` | 16.6 | – |
| | `OctoCoder-15B` | 30.4 | – |
| | `DeepSeekCoder-33B-Instruct` | 47.5 | – |
| | `StarCoder2-15B` | 48.6 | – |
| SWE-agent, 1-shot (Yang et al., 2024) | `gpt-4-turbo` | 87.7 | – |
| OH CodeActAgent v1.5, Generalist, 0-shot. | `gpt-3.5-turbo-16k-0613` | 20.1 | 0.11 |
| | `gpt-4o-2024-05-13` | 79.3 | 0.14 |
| **BIRD** (Li et al., 2023b), 300 instances | | | |
| Prompting, 0-shot | `CodeLlama-7B-Instruct` | 18.3 | - |
| | `CodeQwen-7B-Chat` | 31.3 | - |
| OH CodeActAgent v1.5 | `gpt-4-1106-preview` | 42.7 | 0.19 |
| | `gpt-4o-2024-05-13` | 47.3 | 0.11 |
| **ML-Bench** (Tang et al., 2024b), 68 instances | | | |
| prompting + BM25, 0-shot | `gpt-3.5-turbo` | 11.0 | - |
| | `gpt-4-1106-preview` | 22.1 | - |
| | `gpt-4o-2024-05-13` | 26.2 | - |
| SWE-Agent (Yang et al., 2024) | `gpt-4-1106-preview` | 42.6 | 1.91 |
| Aider (Gauthier) | `gpt-4o` | 64.4 | - |
| OH CodeActAgent v1.5 | `gpt-4o-2024-05-13` | 76.5 | 0.25 |
| | `gpt-4-1106-preview` | 58.8 | 1.22 |
| | `gpt-3.5-turbo-16k-0613` | 13.2 | 0.12 |
| **BioCoder (Python)** (Tang et al., 2024b), 157 instances | | | |
| prompting, 0-shot | `gpt-3.5-turbo` | 11.0 | - |
| | `gpt-4-1106-preview` | 12.7 | - |
| OH CodeActAgent v1.5 | `gpt-4o-2024-05-13` | 27.5 | 0.13 |
| **Gorilla APIBench** (Patil et al., 2023), 1775 instances | | | |
| Prompting, 0-shot | `claude-v1` | 8.7 | - |
| | `gpt-4-0314` | 21.2 | - |
| | `gpt-3.5-turbo-0301` | 29.7 | - |
| Gorilla, finetuned for API calls, 0-shot (Patil et al., 2023; Touvron et al., 2023) | `llama-7b` | 75.0 | - |
| OH CodeActAgent v1.5 | `gpt-3.5-turbo-0125` | 21.6 | 0.002 |
| | `gpt-4o-2024-05-13` | 36.4 | 0.04 |
| **ToolQA** (Zhuang et al., 2024), 800 instances | | | |
| Prompting, 0-shot | `ChatGPT + CoT` | 5.1 | - |
| | `ChatGPT` | 5.6 | - |
| | `Chameleon` | 10.6 | - |
| ReAct, 0-shot (Yao et al., 2023; OpenAI, 2024a) | `gpt-3.5-turbo` | 36.8 | - |
| | `gpt-3` | 43.1 | - |
| OH CodeActAgent v1.5 | `gpt-3.5-turbo-0125` | 2.3 | 0.03 |
| | `gpt-4o-2024-05-13` | 47.2 | 0.91 |

carefully validated, achieving 100% on this benchmark is entirely feasible, which we seek to do in future iterations of OpenHands.

**ML-Bench** (Tang et al., 2024b) evaluates agents' ability to solve machine learning tasks across 18 GitHub repositories. The benchmark comprises 9,641 tasks spanning 169 diverse ML problems, requiring agents to generate bash scripts or Python code in response to user instructions. In the sandbox environment, agents can iteratively execute commands and receive feedback, allowing them to understand the repository context and fulfill user requirements progressively. Following the setup from the original paper, we perform agent evaluation on the quarter subset of ML-Bench.

**Gorilla APIBench** (Patil et al., 2023) evaluates agents' abilities to use APIs. it incorporates tasks on TorchHub, TensorHub, and HuggingFace. During the evaluation, models are given a question related to API usage, such as "*identify an API capable of converting spoken language in a recording to text.*" Correctness is evaluated based on whether the model's API call is in the correct domain.

**ToolQA** (Zhuang et al., 2024) evaluates agents' abilities to use external tools. This benchmark includes tasks on various topics like flight status, coffee price, Yelp data, and Airbnb data, requiring the use of various tools such as text tools, database tools, math tools, graph tools, code tools, and system tools. It features two levels: easy and hard. Easy questions focus more on single-tool usage, while hard questions emphasize reasoning. We adopt the easy subset for evaluation.

**BioCoder** (Tang et al., 2024c) is a repository-level code generation benchmark that evaluates agents' performance on bioinformatics-related tasks, specifically the ability to retrieve and accurately utilize context. The original prompts contain the relevant context of the code; however, in this study, we have removed them to demonstrate the capability of OpenHands to perform context retrieval,

Table 5: OpenHands Web Browsing Evaluation Results (§4.3).

| Agent | Model | Success Rate (%) | $ Avg. Cost |
|---|---|---|---|
| **WebArena** (Zhou et al., 2023a), 812 instances | | | |
| Lemur (Xu et al., 2023) | Lemur-chat-70b | 5.3 | – |
| Patel et al. (2024) | Trained 72B with self-improvement synthetic data | 9.4 | – |
| AutoWebGLM (Lai et al., 2024) | Trained 7B with human/agent hybrid annotation | 18.2 | – |
| Auto Eval & Refine (Pan et al., 2024) | GPT-4 + Reflexion w/ GPT-4V reward model | 20.2 | – |
| WebArena Agent (Zhou et al., 2023a) | gpt-3.5-turbo | 6.2 | – |
| | gpt-4-turbo | 14.4 | – |
| OH BrowsingAgent v1.0 | gpt-4o-mini-2024-07-18 | 8.5 | 0.01 |
| | gpt-4o-2024-05-13 | 14.8 | 0.15 |
| | claude-3-5-sonnet-20240620 | 15.5 | 0.10 |
| OH CodeActAgent v1.8 via **delegation** to BrowsingAgent v1.0 | gpt-4o-mini-2024-07-18 | 8.3 | – |
| | gpt-4o-2024-05-13 | 14.5 | – |
| | claude-3-5-sonnet-20240620 | 15.3 | – |
| **MiniWoB++** (Liu et al., 2018), 125 environments | | | |
| Workflow Guided Exploration (Liu et al., 2018) | Trained specialist model with environment exploration | 34.6 | – |
| CC-NET (Humphreys et al., 2022) | Trained specialist model with RL and human annotated BC | 91.1 | – |
| OH BrowsingAgent v1.0 | gpt-3.5-turbo-0125 | 27.2 | 0.01 |
| | gpt-4o-2024-05-13 | 40.8 | 0.05 |
| OH CodeActAgent v1.8 via **delegation** to BrowsingAgent v1.0 | gpt-4o-2024-05-13 | 39.8 | – |

self-debugging, and reasoning in multi-turn interactions. BioCoder consists of 157 Python and 50 Java functions, each targeting a specific area in bioinformatics, such as proteomics, genomics, and other specialized domains. The benchmark targets real-world code by generating code in existing repositories where the relevant code has been masked out.

**BIRD** (Li et al., 2023b) is a benchmark for text-to-SQL tasks (*i.e.*, translate natural language into executable SQL) aimed at realistic and large-scale database environments. We select 300 samples from the dev set to integrate into OpenHands and evaluate on execution accuracy. Additionally, we extend the setting by allowing the agent to engage in multi-turn interactions to arrive at the final SQL query, enabling it to correct historical results by observing the results of SQL execution.

## 4.3 WEB BROWSING

We report evaluation results for web browsing benchmarks in Tab. 5.

**WebArena** (Zhou et al., 2023a) is a self-hostable, execution-based web agent benchmark that allows agents to freely choose which path to take in completing their given tasks. WebArena comprises 812 human-curated task instructions across various domains, including shopping, forums, developer platforms, and content management systems. **Results.** From Tab. 5, we can see that our BrowsingAgent achieves competitive performance among agents that use LLMs with domain-general prompting techniques.

**MiniWoB++** (Liu et al., 2018) is an interactive web benchmark, with built-in reward functions. The tasks are synthetically initialized on 125 different minimalist web interfaces. Unlike WebArena, tasks are easier without page changes, require fewer steps, and provide low-level step-by-step task directions. Note that it contains a portion of environments that require vision capability to tackle successfully, and many existing work choose to focus only on a subset of the tasks (Kim et al., 2024; Li et al., 2023d; Shaw et al., 2023). Still, we report the performance on the full set and only include baselines that are evaluated on the full set.

## 4.4 MISCELLANEOUS ASSISTANCE

Results for miscellaneous assistance benchmarks are reported in Tab. 6.

**GAIA** (Mialon et al., 2023) evaluates agents' general task-solving skills, covering different real-world scenarios. It requires various agent capabilities, including reasoning, multi-modal understanding, web browsing, and coding. GAIA consists of 466 curated tasks across three levels. Setting up GAIA is traditionally challenging due to the complexity of integrating various tools with the agent, but OpenHands's infrastructure (*e.g.*, runtime §2.2, tools §2.3) simplifies the integration significantly.

**GPQA** (Rein et al., 2023) evaluates agents' ability for coordinated tool use when solving challenging graduate-level problems. Tool use (*e.g.*, python) and web search are often useful to assist agents in answering these questions since they provide accurate calculations that LLMs are often incapable of and access to information outside of the LLM's parametric knowledge base.

Table 6: OpenHands miscellaneous assistance evaluation results (§4.4).

| Agent | Model | Success Rate (%) | $ Avg. Cost |
|---|---|---|---|
| **GAIA** (Mialon et al., 2023), L1 validation set, 53 instances | | | |
| AutoGPT (Gravitas, 2023) | `gpt-4-turbo` | 13.2 | – |
| OH GPTSwarm v1.0 | `gpt-4-0125-preview` | 30.2 | 0.110 |
| | `gpt-4o-2024-05-13` | 32.1 | 0.050 |
| **GPQA** (Rein et al., 2023), diamond set, 198 instances (refer to §G, Tab. 7 for other subsets) | | | |
| Human (Rein et al., 2023) | Expert human | 81.3 | – |
| | Non-expert human | 21.9 | – |
| Few-shot Prompting + Chain-of-Thought (Rein et al., 2023) | `gpt-3.5-turbo-16k` | 29.6 | – |
| | `gpt-4` | 38.8 | – |
| OH CodeActAgent v1.8 | `claude-3-5-sonnet-20240620` | 52.0 | 0.065 |
| **AgentBench** (Liu et al., 2023), OS (bash) subset, 144 instances | | | |
| AgentBench Baseline Agent (Liu et al., 2023) | `gpt-4` | 42.4 | – |
| | `gpt-3.5-turbo` | 32.6 | – |
| OH CodeActAgent v1.5 | `gpt-4o-2024-05-13` | 57.6 | 0.085 |
| | `gpt-3.5-turbo-0125` | 11.8 | 0.006 |
| **MINT** (Wang et al., 2024b): `math` subset, 225 instances | | | |
| MINT Baseline Agent | `gpt-4-0613` | 65.8 | – |
| OH CodeActAgent v1.5 | `gpt-4o-2024-05-13` | 77.3 | 0.070 |
| | `gpt-3.5-turbo-16k-0613` | 33.8 | 0.048 |
| **MINT** (Wang et al., 2024b): `code` subset, 136 instances | | | |
| MINT Baseline Agent | `gpt-4-0613` | 59.6 | – |
| OH CodeActAgent v1.5 | `gpt-4o-2024-05-13` | 50.0 | 0.087 |
| | `gpt-3.5-turbo-16k-0613` | 5.2 | 0.030 |
| **ProofWriter** (Tafjord et al., 2021), 600 instances | | | |
| Few-shot Prompting + Chain-of-Thought (Pan et al., 2023) | `gpt4` | 68.1 | – |
| Logic-LM (Pan et al., 2023) | `gpt4 + symbolic solver` | 79.6 | – |
| OH CodeActAgent v1.5 | `gpt-4o-2024-05-13` | 78.8 | – |
| **Entity Deduction Arena** (Zhang et al., 2024a), 200 instances | | | |
| Zero-shot Prompting (Zhang et al., 2024a) | `gpt-4-0314` | 40.0 | – |
| | `gpt-3.5-turbo-0613` | 27.0 | – |
| OH CodeActAgent v1.5 | `gpt-4o-2024-05-13` | 38.0 | – |
| | `gpt-3.5-turbo-16k-0613` | 24.0 | – |

**AgentBench** (Liu et al., 2023) evaluates agents' reasoning and decision-making abilities in a multi-turn, open-ended generation setting. We selected the code-grounded operating system (OS) subset with 144 tasks. Agents from OpenHands interact directly with the task-specific OS using bash commands in a multi-turn manner, combining interaction and reasoning to automate task completion.

**MINT** (Wang et al., 2024b) is a benchmark designed to evaluate agents' ability to solve challenging tasks through *multi-turn interactions* using *tools* and *natural language feedback* simulated by GPT-4. We use coding and math subsets used in Yuan et al. (2024). We follow the original paper and allow the agent to interact with up to five iterations with two chances to propose solutions.

**ProofWriter** (Tafjord et al., 2021) is a synthetic dataset created to assess deductive reasoning abilities of LLMs. Same as Logic-LM (Pan et al., 2023), we focus on the most challenging subset, which contains 600 instances requiring 5-hop reasoning. To minimize the impact of potential errors in semantic parsing, we use the logical forms provided by Logic-LM.

**Entity Deduction Arena** (EDA) (Zhang et al., 2024a) evaluates agents' ability to deduce unknown entities through strategic questioning, akin to the 20 Questions game. This benchmark tests the agent's state tracking, strategic planning, and inductive reasoning capabilities over multi-turn conversations. We evaluate two datasets "Things" and "Celebrities", each comprising 100 instances, and report the average success rate over these two datasets.

## 5 CONCLUSION

We introduce OpenHands, a community-driven platform that enables the development of agents that interact with the world through software interfaces. By providing a powerful interaction mechanism, a safe sandboxed environment, essential agent skills, multi-agent collaboration capabilities, and a comprehensive evaluation framework, OpenHands accelerates research innovations and real-world applications of agentic AI systems. Despite challenges in developing safe and reliable agents (§A), we are excited about our vibrant community and look forward to OpenHands's continued evolution.

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

AUTHOR CONTRIBUTIONS

This work was an open-source collaborative effort across multiple institutions. We employed a point-based system to determine contributions and award authorships, with technical contributions tracked and measured in units of pull requests (PRs)[3]. Xingyao Wang led the project, coordinating overall development and paper writing efforts. Detailed contributions were as follows:

- **Agent Development** (§3): Xingyao Wang led the implementation of CodeAct Wang et al. (2024a) and CodeActSWE agents. Frank F. Xu led the development of web browsing agents Zhou et al. (2023a). Mingchen Zhuge orchestrated the integration of the GPTSwarm agent Zhuge et al. (2024). Robert Brennan and Boxuan Li lead the development of the Micro Agent.

- **Architectural Development** (Fig. 2): Robert Brennan initiated the architecture design. Boxuan Li, Frank F. Xu, Xingyao Wang, Yufan Song, and Mingzhang Zheng further refined and expanded the architecture. Boxuan Li implemented the initial version of integration tests (§E), maintained the agentskills library (§2.3), managed configurations, and resolved resource leaks in evaluation. Frank F. Xu developed the web browsing environment (§J) for both agent execution and evaluation and integrated it with both agent and front-end user interfaces. Xingyao Wang authored the initial code for the agentskills library and the Docker sandbox. Yufan Song implemented cost tracking for evaluation, while Mingzhang Zheng developed an image-agnostic docker sandbox for more stable SWE-Bench evaluation.

- **Benchmarking, Integration, and Code Review**: Boxuan Li and Yufan Song led benchmark integration efforts, including coordination, evaluation, and code review. Yufan Song also helped track PR contributions. Graham Neubig, Xingyao Wang, Mingzhang Zheng, Robert Brennan, Hoang H. Tran, Frank F. Xu, Xiangru Tang, Fuqiang Li, and Yanjun Shao provided additional support in integration and code reviews. Specific benchmark contributions included:
  - SWE-Bench Jimenez et al. (2024): Bowen Li and Xingyao Wang
  - WebArena Zhou et al. (2023a) and MiniWob++ Liu et al. (2018): Frank F. Xu
  - GAIA Mialon et al. (2023): Jiayi Pan (integration) and Mingchen Zhuge (GPTSwarm evaluation)
  - API-Bench Patil et al. (2023) and ToolQA Zhuang et al. (2024): Yueqi Song
  - HumanEvalFix Muennighoff et al. (2024): Niklas Muennighoff and Xiangru Tang
  - ProofWriter Tafjord et al. (2021): Ren Ma
  - MINT Wang et al. (2024b): Hoang H. Tran
  - AgentBench Liu et al. (2023): Fuqiang Li
  - BIRD Li et al. (2023b): Binyuan Hui
  - GPQA Rein et al. (2023): Jaskirat Singh
  - BioCoder Tang et al. (2024c): Xiangru Tang and Bill Qian
  - ML-Bench Tang et al. (2024b): Xiangru Tang and Yanjun Shao
  - Entity-Deduction-Arena Zhang et al. (2024a): Yizhe Zhang

- **Advising**: Graham Neubig advised the project, providing guidance, resources, and substantial paper edits. Heng Ji and Hao Peng offered additional project advice and assisted with paper writing. Junyang Lin contributed advisory support and sponsored resources.

## A   LIMITATIONS AND FUTURE WORK

We are excited about the foundations our vibrant community has laid in OpenHands and look forward to its continued evolution. We identify several directions for future work:

**Enhanced multi-modality support.** While our current implementation already supports a wide range of file formats through predefined agent skills, we are interested in enabling multi-modality in a principled way through standard IPython and browser integration, such as viewing images and videos using vision-language model through a browser or processing XLSX files with code.

**Stronger agents.** Current agents still struggle with complex tasks, and we are interested in building better agents through both training and inference time techniques.

---

[3]For more details, please refer to https://github.com/All-Hands-AI/OpenHands/pull/1917.

**Agent editing improvements.** Current agent suffers a lot when editing long files, and we are interested in exploring different approaches to improve the file editing performance of agents.

**Web browsing improvements.** Due to the extensible nature of OpenHands, orthogonal components that could improve agents can be integrated easily. For example, thanks to OpenHands's extensible architecture, Auto Eval & Refine Pan et al. (2024), an agent retry-on-error strategy with Reflexion Shinn et al. (2024) prompts and task completion reward models, will be integrated as an optional component attached to our browsing agent.

**Automatic workflow generation.** Currently, OpenHands's workflow still requires a substantial hand-crafted workload. We believe that graph-based frameworks such as GPTSwarm Zhuge et al. (2024) and LangGraph Chase (2022) could serve as alternative solutions for building agents. Particularly in GPTSwarm, when agents are constructed using graphs, it becomes easier to incorporate various optimization methods (e.g., reinforcement learning, meta-prompting). OpenHands considers these methods to lay the groundwork for promising solutions in automatic workflow generation in future versions.

## B  ETHICS STATEMENT

Most AI agents today are still research artifacts and lack the ability to perform complex, long-horizon tasks in the real world reliably. However, as their performance continues to improve and they are increasingly deployed in real world, they have the potential to boost productivity while also posing security risks to society significantly. OpenHands helps mitigate risks by:

(1) Enabling systematic evaluation of these agents, which can identify and address risks before they are widely deployed.

(2) Facilitating human-agent interaction rather than allowing agents to operate autonomously without oversight.

(3) More importantly, we hope OpenHands allows researchers worldwide to access the best suites of agents to conduct frontier safety research towards building safe and helpful agents.

## C  RELATED WORK

The breakthroughs in large language models (LLMs) like ChatGPT OpenAI (2024a) and GPT-4 OpenAI et al. (2024) have significantly enhanced the capabilities of autonomous agents across various domains Ye et al. (2023); Tang et al. (2024d); Park et al. (2023); Cui et al. (2023). These advances have spurred a multitude of generalist agent proposals Gravitas (2023); Nakajima (2023); Wu et al. (2023) aimed at performing diverse user tasks and have gained attention from both developers and broader audiences. Notable works such as Auto-GPT Gravitas (2023) harness LLMs for task completion by decomposing user goals into executable steps. Multi-agent collaboration systems leverage LLMs for elements like role-playing and task-solving capabilities Zhuge et al. (2023); Li et al. (2023a); Zhou et al. (2023b); Team (2023), with MetaGPT Hong et al. (2023) emphasizing standardized operating procedures, and AutoGen Wu et al. (2023) providing a conversation framework for interactive systems. AGENTS Zhou et al. (2023b) and AutoAgents Chen et al. (2024) offer new paradigms for customizable agent architecture, while XAgent Team (2023) and GPTSwarm Zhuge et al. (2024) introduce complex management systems and optimizable graphs, respectively, for enhanced agent operations.

This surge in agent development has led to specialized frameworks aimed at streamlining agent implementation. LangChain and LangGraph Chase (2022) provide foundational building blocks with basic runtime support, while CrewAI CrewAI (2024) focuses on orchestrating multi-agent communications. BrowserGym ServiceNow specifically targets web browsing capabilities, and DSPy Khattab et al. (2024) emphasizes end-to-end prompt optimization. AutoGen Wu et al. (2023) advances beyond basic frameworks by implementing Python and bash execution capabilities, though with stateless command execution, while frameworks like CrewAI offer sandboxed but limited code interpreter features.

Software development, a front-runner in applying LLM-based agents, has seen advancements in frameworks for facilitating the development processes Hong et al. (2023); Qian et al. (2023). In-

novations such as ChatDev Qian et al. (2023) automate the software development lifecycle akin to the waterfall model, and AutoCodeRover Zhang et al. (2024b) addresses GitHub issues via code search and abstract syntax tree manipulation. AgentCoder Huang et al. (2024) iteratively refines code generation with integrated testing and feedback, while SWE-Agent Yang et al. (2024) integrates LLMs for automated Github issue fixing, streamlining software engineering.

## D    GRAPHICAL USER INTERFACE

Besides running from the command line, OpenHands features a rich graphical user interface that visualizes the agent's current actions (*e.g.*, browsing the web, executing base commands or Python code, *etc*.) and allows for real-time feedback from the user. Screenshots of the UI are shown in Fig. 1. The user may interrupt the agent at any moment to provide additional feedback, comments, or instruction while the agent is working. This user interface directly connects with the event streams (§2.1) to control and visualize the agents and runtime, making it agent and runtime agnostic.

## E    QUALITY CONTROL: INTEGRATION TESTS FOR AGENTS

Integration tests Leung & White (1990) have long been used by software developers to ensure software quality. Unlike large language models with simple input-output schema, agents are typically complex pieces of software where minor errors can be easily introduced during the development process and hurt final task performance. While running a full suite evaluation (§4) is the ultimate measure of performance degradation, running them for *every* code changes can be prohibitively slow and expensive. [4]. In OpenHands, we pioneer an end-to-end agent test framework that tests prompt regression, actions, and sandbox environments. It combines integration testing from software engineering and foundation model mocking for deterministic behavior to prevent the accidental introduction of bugs during agent development.

**Defining an integration test.** The integration test framework for OpenHands is structured to validate end-to-end functionality by automating task execution and result verification. Developers define tasks and expected results; for instance, a task might involve correcting typos in a document named "bad.txt". Upon task execution through OpenHands, outputs are compared against a predefined "gold file" to ensure accuracy.

**Mocking LLM for deterministic behavior.** Addressing the challenge of non-determinism in large language models (LLMs) and the associated high costs, the framework intercepts all LLM calls and supplies predefined responses based on exact prompt matches. This method not only ensures consistency in test outcomes but also reduces operational costs by minimizing the reliance on real LLMs.

**Regenerate LLM responses on breaking changes.** Prompt-response pairs are managed through a script that generates and stores these pairs when new tests are introduced or existing prompts are modified. For routine tests, the framework attempts to reuse existing LLM responses by slightly adjusting the prompts. Substantial changes that affect task handling require regeneration of these pairs using real LLMs.

**Benefits of integration tests.** The framework offers several advantages, including 1) Prompt regression testing: Stored prompt-response pairs facilitate change tracking and provide a reference for new team members to understand LLM interactions, 2) Multi-platform support: Tests are automatically scheduled for every pull request and commit on the main branch, running across multiple platforms, environments, and agents, including Linux and Mac, and in local, SSH, and exec sandboxes, and 3) Comprehensive error detection: It captures errors in prompt generation, message passing, and sandbox execution, thereby maintaining a high test coverage.

---

[4]Running a SWE-Bench Lite Jimenez et al. (2024) evaluation with `gpt-4o` costs around 600 USD.

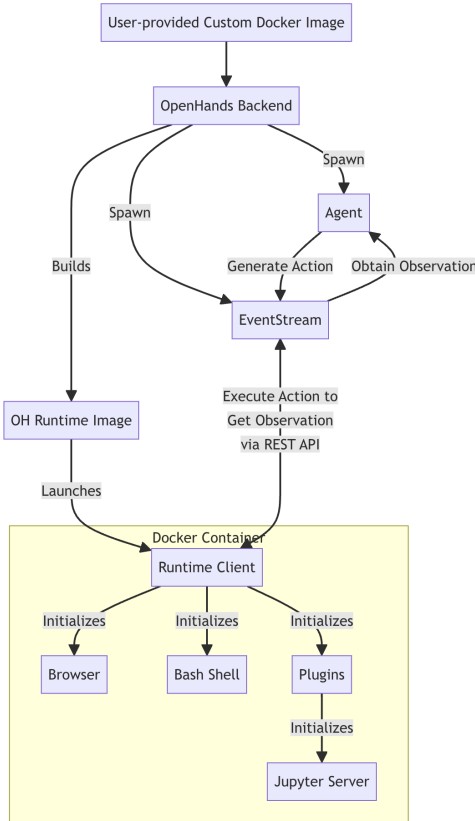

Figure 4: OpenHands runtime workflow.

## F   HOW OPENHANDS RUNTIME WORK

### F.1   WORKFLOW

The OpenHands Runtime system uses a client-server architecture implemented with Docker containers. See Fig. 4 for an overview of how it works.

(1) **User Input**: The user provides a custom base Docker image.

(2) **Image Building**: OpenHands builds a new Docker image (the "OH runtime image") based on the user-provided image. This new image includes OpenHands-specific code, primarily the "runtime client" (i.e., runtime API server described in §2.2).

(3) **Container Launch**: When OpenHands starts, it launches a Docker container using the OH runtime image.

(4) **Communication**: The OpenHands backend (`runtime.py`) communicates with the runtime client over RESTful API, sending actions and receiving observations

(5) **Action Execution**: The runtime client receives actions from the backend, executes them in the sandboxed environment, and sends back observations

(6) **Observation Return**: The client sends execution results back to the OpenHands backend event stream as observations.

The role of the client:

- It acts as an intermediary between the OpenHands backend and the sandboxed environment

- It executes various types of actions (shell commands, file operations, Python code, etc.) safely within the container

- It manages the state of the sandboxed environment, including the current working directory and loaded plugins

- It formats and returns observations to the backend, ensuring a consistent interface for processing results

## F.2 HOW OPENHANDS BUILDS AND MAINTAINS RUNTIME IMAGES

OpenHands' approach to building and managing runtime images ensures efficiency, consistency, and flexibility in creating and maintaining Docker images for both production and development environments.

### F.2.1 IMAGE TAGGING SYSTEM

OpenHands uses a dual-tagging system for its runtime images to balance reproducibility with flexibility:

(1) Hash-based tag: `{target_image_repo}:{target_image_hash_tag}`. Example: `runtime:abc123def456`

- This tag is based on the MD5 hash of the Docker build folder, which includes the source code (of runtime client and related dependencies) and Dockerfile
- Identical hash tags guarantee that the images were built with exactly the same source code and Dockerfile
- This ensures reproducibility; the same hash always means the same image contents

(2) Generic tag: `{target_image_repo}:{target_image_tag}`. Example: `runtime:oh_v0.9.3_ubuntu_tag_22.04`

- This tag follows the format: `runtime:oh_v{VERSION}_{BASE_IMAGE}_tag_{IMAGE_TAG}`
- It represents the latest build for a particular base image and OpenHands version combination
- This tag is updated whenever a new image is built from the same base image, even if the source code changes

The hash-based tag ensures reproducibility, while the generic tag provides a stable reference to the latest version of a particular configuration. This dual-tagging approach allows OpenHands to efficiently manage both development and production environments.

### F.2.2 BUILD PROCESS

(1) **Image Naming Convention**:

- Hash-based tag: `target_image_repo:target_image_hash_tag`. Example: `runtime:abc123def456`
- Generic tag: `target_image_repo:target_image_tag`. Example: `runtime:oh_v0.9.3_ubuntu_tag_22.04`

(2) **Build Process**:

a. Convert the base image name to an OH runtime image name Example: `ubuntu:22.04` -> `runtime:oh_v0.9.3_ubuntu_tag_22.04`
b. Generate a build context (Dockerfile and OpenHands source code) and calculate its hash
c. Check for an existing image with the calculated hash
d. If not found, check for a recent compatible image to use as a base
e. If no compatible image exists, build from scratch using the original base image
f. Tag the new image with both hash-based and generic tags

(3) **Image Reuse and Rebuilding Logic**: The system follows these steps to determine whether to build a new image or use an existing one from a user-provided (base) image (e.g., `ubuntu:22.04`):

a. If an image exists with the same hash (e.g., `runtime:abc123def456`), it will be reused as is
b. If the exact hash is not found, the system will try to rebuild using the latest generic image (e.g., `runtime:oh_v0.9.3_ubuntu_tag_22.04`) as a base. This saves time by leveraging existing dependencies

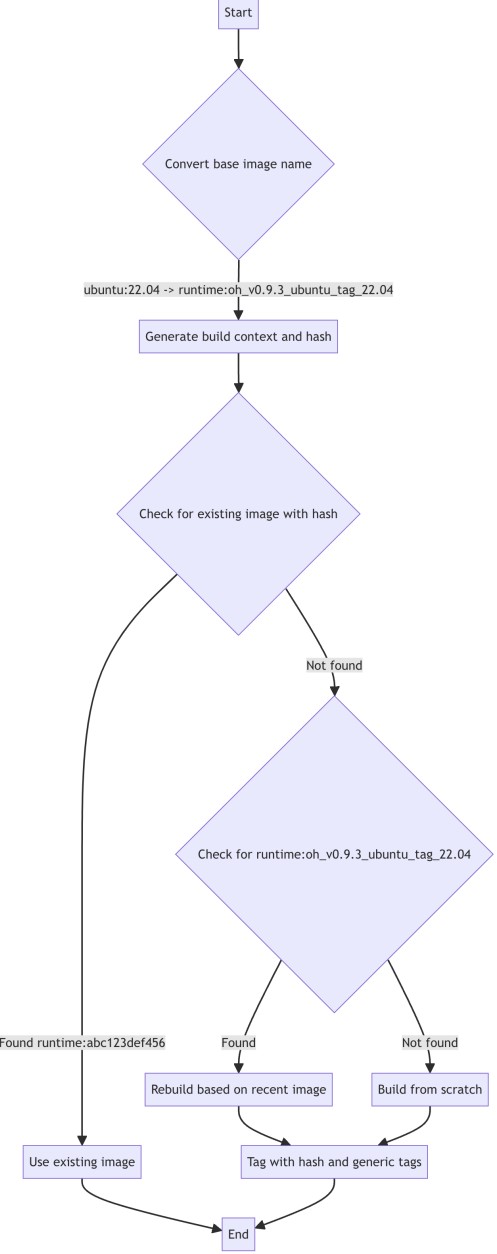

Figure 5: OpenHands Runtime Image Build Workflow.

    c. If neither the hash-tagged nor the generic-tagged image is found, the system will build the image completely from scratch

**Caching and Efficiency.** The system attempts to reuse existing images when possible to save build time. If an exact match (by hash) is found, it's used without rebuilding. If a compatible image is found, it's used as a base for rebuilding, saving time on dependency installation.

A flowchart illustrating the build process is shown in Fig. 5

## G  ADDITIONAL RESULTS FOR GPQA BENCHMARK

We showcase more detailed results, including performance on other subsets for GPQA benchmark in Tab. 7.

Table 7: *Full Evaluation Results on the GPQA Benchmark Rein et al. (2023) (§4.4).*

| Evaluation Method and Model | Accuracy by subset (%) | | | Avg Cost ($) |
| --- | --- | --- | --- | --- |
| | *Diamond Set* | *Main Set* | *Extended Set* | |
| Expert Human Validators | 81.2 | 72.5 | 65.4 | N/A |
| Non-Expert Human Validators | 21.9 | 30.5 | 33.9 | N/A |
| Few-Shot CoT Llama-2-70B-chat | 28.1 | 29.1 | 30.4 | N/A |
| Few-Shot CoT GPT-3.5-turbo-16k | 29.6 | 28.0 | 28.2 | N/A |
| Few-Shot CoT GPT-4 | 38.8 | 39.7 | 38.7 | N/A |
| GPT-4 with search (backoff to CoT on abstention) | 38.8 | 41.0 | 39.4 | N/A |
| OpenHands + CodeActAgent v1.5 + GPT3.5-turbo | 27.9 | 23.4 | 26.1 | 0.012 |
| OpenHands + CodeActAgent v1.5 + GPT4-turbo | 51.8 | 47.4 | 42.4 | 0.501 |
| OpenHands + CodeActAgent v1.5 + GPT4o | **53.1** | **49.3** | **52.8** | 0.054 |

## H  IN-CONTEXT DEMONSTRATION FOR CODEACTSWEAGENT

The prompt is re-adopted from the SWE-agent's released trajectory (https://github.com/princeton-nlp/SWE-agent/tree/main/trajectories/demonstrations). The prompt can be found at https://github.com/All-Hands-AI/OpenHands/blob/main/agenthub/codeact_swe_agent/prompt.py.

## I  SUPPORTED AGENTSKILLS

As of OpenHands v0.6, we support the following list of skills. Please refer to the source code for the most up-to-date list of skills: https://github.com/All-Hands-AI/OpenHands/blob/main/OpenHands/runtime/plugins/agent_skills/agentskills.py

```python
def open_file(path: str, line_number: Optional[int] = None) ->
↪   None:
    """
    Opens the file at the given path in the editor. If line_number
    ↪   is provided, the window will be moved to include that
    ↪   line.

    Args:
        path: str: The path to the file to open.
        line_number: Optional[int]: The line number to move to.
    """
    pass

def goto_line(line_number: int) -> None:
    """
    Moves the window to show the specified line number.

    Args:
        line_number: int: The line number to move to.
    """
    pass

def scroll_down() -> None:
    """Moves the window down by 100 lines.

    Args:
        None
    """
    pass
```

```python
def scroll_up() -> None:
    """Moves the window up by 100 lines.

    Args:
        None
    """
    pass

def create_file(filename: str) -> None:
    """Creates and opens a new file with the given name.

    Args:
        filename: str: The name of the file to create.
    """
    pass

def edit_file(start: int, end: int, content: str) -> None:
    """Edit a file.

    It replaces lines `start` through `end` (inclusive) with the
    ↪  given text `content` in the open file. Remember, the file
    ↪  must be open before editing.

    Args:
        start: int: The start line number. Must satisfy start >=
        ↪  1.
        end: int: The end line number. Must satisfy start <= end
        ↪  <= number of lines in the file.
        content: str: The content to replace the lines with.
    """
    pass

def search_dir(search_term: str, dir_path: str = './') -> None:
    """Searches for search_term in all files in dir. If dir is not
    ↪  provided, searches in the current directory.

    Args:
        search_term: str: The term to search for.
        dir_path: Optional[str]: The path to the directory to
        ↪  search.
    """
    pass

def search_file(search_term: str, file_path: Optional[str] = None)
↪  -> None:
    """Searches for search_term in file. If file is not provided,
    ↪  searches in the current open file.

    Args:
        search_term: str: The term to search for.
        file_path: Optional[str]: The path to the file to search.
    """
    pass

def find_file(file_name: str, dir_path: str = './') -> None:
    """Finds all files with the given name in the specified
    ↪  directory.

    Args:
```

```python
        file_name: str: The name of the file to find.
        dir_path: Optional[str]: The path to the directory to
        ↪  search.
    """
    pass

def parse_pdf(file_path: str) -> None:
    """Parses the content of a PDF file and prints it.

    Args:
        file_path: str: The path to the file to open.
    """
    pass

def parse_docx(file_path: str) -> None:
    """
    Parses the content of a DOCX file and prints it.

    Args:
        file_path: str: The path to the file to open.
    """
    pass

def parse_latex(file_path: str) -> None:
    """
    Parses the content of a LaTex file and prints it.

    Args:
        file_path: str: The path to the file to open.
    """
    pass

def parse_audio(file_path: str, model: str = 'whisper-1') -> None:
    """
    Parses the content of an audio file and prints it.

    Args:
        file_path: str: The path to the audio file to transcribe.
        model: Optional[str]: The audio model to use for
        ↪  transcription. Defaults to 'whisper-1'.
    """
    pass

def parse_image(
    file_path: str, task: str = 'Describe this image as detail as
    ↪  possible.'
) -> None:
    """
    Parses the content of an image file and prints the
    ↪  description.

    Args:
        file_path: str: The path to the file to open.
        task: Optional[str]: The task description for the API
        ↪  call. Defaults to 'Describe this image as detail as
        ↪  possible.'.
    """
    pass
```

```python
def parse_video(
    file_path: str,
    task: str = 'Describe this image as detail as possible.',
    frame_interval: int = 30,
) -> None:
    """
    Parses the content of an image file and prints the
    ↪   description.

    Args:
        file_path: str: The path to the video file to open.
        task: Optional[str]: The task description for the API
        ↪   call. Defaults to 'Describe this image as detail as
        ↪   possible.'.
        frame_interval: Optional[int]: The interval between frames
        ↪   to analyze. Defaults to 30.

    """
    pass

def parse_pptx(file_path: str) -> None:
    """
    Parses the content of a pptx file and prints it.

    Args:
        file_path: str: The path to the file to open.
    """
    pass
```

## J  BROWSERGYM ACTIONS

The following are all the supported actions defined in BrowserGym[5] as of v0.3.4. The actions can be categorized into several types and can be configured to use only a subset of the functionality. There are agent control actions, navigation actions, page element-based actions, coordinate-based actions, as well as tab-related actions. We use these actions from the BrowserGym library as our main browsing action primitives.

```python
def send_msg_to_user(text: str):
    """
    Sends a message to the user.

    Examples:
        send_msg_to_user("Based on the results of my search, the
        ↪   city was built in 1751.")
    """
    pass

def report_infeasible(reason: str):
    """
    Notifies the user that their instructions are infeasible.

    Examples:
        report_infeasible("I cannot follow these instructions
        ↪   because there is no email field in this form.")
    """
```

---

[5]https://github.com/ServiceNow/BrowserGym/blob/main/core/src/browsergym/core/action/functions.py

```python
        pass

    def noop(wait_ms: float = 1000):
        """
        Do nothing, and optionally wait for the given time (in
        ↪   milliseconds).

        Examples:
            noop()
            noop(500)
        """
        pass

    # https://playwright.dev/docs/input#text-input
    def fill(bid: str, value: str):
        """
        Fill out a form field. It focuses the element and triggers an
        ↪   input event with the entered text.
        It works for <input>, <textarea> and [contenteditable]
        ↪   elements.

        Examples:
            fill('237', 'example value')
            fill('45', "multi-line\\nexample")
            fill('a12', "example with \\"quotes\\"")
        """
        pass

    #
    ↪   https://playwright.dev/python/docs/api/class-locator#locator-check
    def check(bid: str):
        """
        Ensure a checkbox or radio element is checked.

        Examples:
            check('55')
        """
        pass

    #
    ↪   https://playwright.dev/python/docs/api/class-locator#locator-uncheck
    def uncheck(bid: str):
        """
        Ensure a checkbox or radio element is unchecked.

        Examples:
            uncheck('a5289')
        """
        pass

    # https://playwright.dev/docs/input#select-options
    def select_option(bid: str, options: str | list[str]):
        """
```

```
        Select one or multiple options in a <select> element. You can
        ↪   specify
        option value or label to select. Multiple options can be
        ↪   selected.

        Examples:
            select_option('a48', "blue")
            select_option('c48', ["red", "green", "blue"])
        """
        pass

    #
    ↪   https://playwright.dev/python/docs/api/class-locator#locator-click
    def click(
        bid: str,
        button: Literal["left", "middle", "right"] = "left",
        modifiers: list[Literal["Alt", "Control", "Meta", "Shift"]] =
        ↪   [],
    ):
        """
        Click an element.

        Examples:
            click('a51')
            click('b22', button="right")
            click('48', button="middle", modifiers=["Shift"])
        """
        pass

    #
    ↪   https://playwright.dev/python/docs/api/class-locator#locator-dblclick
    def dblclick(
        bid: str,
        button: Literal["left", "middle", "right"] = "left",
        modifiers: list[Literal["Alt", "Control", "Meta", "Shift"]] =
        ↪   [],
    ):
        """
        Double click an element.

        Examples:
            dblclick('12')
            dblclick('ca42', button="right")
            dblclick('178', button="middle", modifiers=["Shift"])
        """
        pass

    #
    ↪   https://playwright.dev/python/docs/api/class-locator#locator-hover
    def hover(bid: str):
        """
        Hover over an element.

        Examples:
            hover('b8')
        """
```

```python
        pass

# https://playwright.dev/python/docs/input#keys-and-shortcuts
def press(bid: str, key_comb: str):
    """
    Focus the matching element and press a combination of keys. It
    ↪    accepts
    the logical key names that are emitted in the
    ↪    keyboardEvent.key property
    of the keyboard events: Backquote, Minus, Equal, Backslash,
    ↪    Backspace,
    Tab, Delete, Escape, ArrowDown, End, Enter, Home, Insert,
    ↪    PageDown, PageUp,
    ArrowRight, ArrowUp, F1 - F12, Digit0 - Digit9, KeyA - KeyZ,
    ↪    etc. You can
    alternatively specify a single character you'd like to produce
    ↪    such as "a"
    or "#". Following modification shortcuts are also supported:
    ↪    Shift, Control,
    Alt, Meta.

    Examples:
        press('88', 'Backspace')
        press('a26', 'Control+a')
        press('a61', 'Meta+Shift+t')
    """
    pass

#
↪    https://playwright.dev/python/docs/api/class-locator#locator-focus
def focus(bid: str):
    """
    Focus the matching element.

    Examples:
        focus('b455')
    """
    pass

#
↪    https://playwright.dev/python/docs/api/class-locator#locator-clear
def clear(bid: str):
    """
    Clear the input field.

    Examples:
        clear('996')
    """
    pass

# https://playwright.dev/python/docs/input#drag-and-drop
def drag_and_drop(from_bid: str, to_bid: str):
    """
    Perform a drag & drop. Hover the element that will be dragged.
    ↪    Press
```

```python
    left mouse button. Move mouse to the element that will receive
    ↪  the
    drop. Release left mouse button.

    Examples:
        drag_and_drop('56', '498')
    """
    pass

# https://playwright.dev/python/docs/api/class-mouse#mouse-wheel
def scroll(delta_x: float, delta_y: float):
    """
    Scroll horizontally and vertically. Amounts in pixels,
    ↪  positive for right or down scrolling, negative for left or
    ↪  up scrolling. Dispatches a wheel event.

    Examples:
        scroll(0, 200)
        scroll(-50.2, -100.5)
    """
    pass

# https://playwright.dev/python/docs/api/class-mouse#mouse-move
def mouse_move(x: float, y: float):
    """
    Move the mouse to a location. Uses absolute client coordinates
    ↪  in pixels.
    Dispatches a mousemove event.

    Examples:
        mouse_move(65.2, 158.5)
    """
    pass

# https://playwright.dev/python/docs/api/class-mouse#mouse-up
def mouse_up(x: float, y: float, button: Literal["left", "middle",
↪  "right"] = "left"):
    """
    Move the mouse to a location then release a mouse button.
    ↪  Dispatches
    mousemove and mouseup events.

    Examples:
        mouse_up(250, 120)
        mouse_up(47, 252, 'right')
    """
    pass

# https://playwright.dev/python/docs/api/class-mouse#mouse-down
def mouse_down(x: float, y: float, button: Literal["left",
↪  "middle", "right"] = "left"):
    """
    Move the mouse to a location then press and hold a mouse
    ↪  button. Dispatches
    mousemove and mousedown events.
```

```python
    Examples:
        mouse_down(140.2, 580.1)
        mouse_down(458, 254.5, 'middle')
    """
    pass

# https://playwright.dev/python/docs/api/class-mouse#mouse-click
def mouse_click(x: float, y: float, button: Literal["left",
↪   "middle", "right"] = "left"):
    """
    Move the mouse to a location and click a mouse button.
    ↪   Dispatches mousemove,
    mousedown and mouseup events.

    Examples:
        mouse_click(887.2, 68)
        mouse_click(56, 712.56, 'right')
    """
    pass

#
↪   https://playwright.dev/python/docs/api/class-mouse#mouse-dblclick
def mouse_dblclick(x: float, y: float, button: Literal["left",
↪   "middle", "right"] = "left"):
    """
    Move the mouse to a location and double click a mouse button.
    ↪   Dispatches
    mousemove, mousedown and mouseup events.

    Examples:
        mouse_dblclick(5, 236)
        mouse_dblclick(87.5, 354, 'right')
    """
    pass

def mouse_drag_and_drop(from_x: float, from_y: float, to_x: float,
↪   to_y: float):
    """
    Drag and drop from a location to a location. Uses absolute
    ↪   client
    coordinates in pixels. Dispatches mousemove, mousedown and
    ↪   mouseup
    events.

    Examples:
        mouse_drag_and_drop(10.7, 325, 235.6, 24.54)
    """
    pass

#
↪   https://playwright.dev/python/docs/api/class-keyboard#keyboard-press
def keyboard_press(key: str):
    """
```

```
    Press a combination of keys. Accepts the logical key names
    ↪   that are
    emitted in the keyboardEvent.key property of the keyboard
    ↪   events:
    Backquote, Minus, Equal, Backslash, Backspace, Tab, Delete,
    ↪   Escape,
    ArrowDown, End, Enter, Home, Insert, PageDown, PageUp,
    ↪   ArrowRight,
    ArrowUp, F1 - F12, Digit0 - Digit9, KeyA - KeyZ, etc. You can
    alternatively specify a single character you'd like to produce
    ↪   such
    as "a" or "#". Following modification shortcuts are also
    ↪   supported:
    Shift, Control, Alt, Meta.

    Examples:
        keyboard_press('Backspace')
        keyboard_press('Control+a')
        keyboard_press('Meta+Shift+t')
        page.keyboard.press("PageDown")
    """
    pass

#
↪   https://playwright.dev/python/docs/api/class-keyboard#keyboard-up
def keyboard_up(key: str):
    """
    Release a keyboard key. Dispatches a keyup event. Accepts the
    ↪   logical
    key names that are emitted in the keyboardEvent.key property
    ↪   of the
    keyboard events: Backquote, Minus, Equal, Backslash,
    ↪   Backspace, Tab,
    Delete, Escape, ArrowDown, End, Enter, Home, Insert, PageDown,
    ↪   PageUp,
    ArrowRight, ArrowUp, F1 - F12, Digit0 - Digit9, KeyA - KeyZ,
    ↪   etc.
    You can alternatively specify a single character you'd like to
    ↪   produce
    such as "a" or "#".

    Examples:
        keyboard_up('Shift')
        keyboard_up('c')
    """
    pass

#
↪   https://playwright.dev/python/docs/api/class-keyboard#keyboard-down
def keyboard_down(key: str):
    """
    Press and holds a keyboard key. Dispatches a keydown event.
    ↪   Accepts the
    logical key names that are emitted in the keyboardEvent.key
    ↪   property of
    the keyboard events: Backquote, Minus, Equal, Backslash,
    ↪   Backspace, Tab,
```

```python
    Delete, Escape, ArrowDown, End, Enter, Home, Insert, PageDown,
    ↪    PageUp,
    ArrowRight, ArrowUp, F1 - F12, Digit0 - Digit9, KeyA - KeyZ,
    ↪    etc. You can
    alternatively specify a single character such as "a" or "#".

    Examples:
        keyboard_up('Shift')
        keyboard_up('c')
    """
    pass

#
↪  https://playwright.dev/python/docs/api/class-keyboard#keyboard-type
def keyboard_type(text: str):
    """
    Types a string of text through the keyboard. Sends a keydown,
    ↪    keypress/input,
    and keyup event for each character in the text. Modifier keys
    ↪    DO NOT affect
    keyboard_type. Holding down Shift will not type the text in
    ↪    upper case.

    Examples:
        keyboard_type('Hello world!')
    """
    pass

#
↪  https://playwright.dev/python/docs/api/class-keyboard#keyboard-insert-text
def keyboard_insert_text(text: str):
    """
    Insert a string of text in the currently focused element.
    ↪    Dispatches only input
    event, does not emit the keydown, keyup or keypress events.
    ↪    Modifier keys DO NOT
    affect keyboard_insert_text. Holding down Shift will not type
    ↪    the text in upper
    case.

    Examples:
        keyboard_insert_text('Hello world!')
    """
    pass

# https://playwright.dev/python/docs/api/class-page#page-goto
def goto(url: str):
    """
    Navigate to a url.

    Examples:
        goto('http://www.example.com')
    """
    pass
```

```python
# https://playwright.dev/python/docs/api/class-page#page-go-back
def go_back():
    """
    Navigate to the previous page in history.

    Examples:
        go_back()
    """
    pass

#
↪   https://playwright.dev/python/docs/api/class-page#page-go-forward
def go_forward():
    """
    Navigate to the next page in history.

    Examples:
        go_forward()
    """
    pass

#
↪   https://playwright.dev/python/docs/api/class-browsercontext#browser-context-new-p
def new_tab():
    """
    Open a new tab. It will become the active one.

    Examples:
        new_tab()
    """
    global page
    # set the new page as the active page
    page = page.context.new_page()
    # trigger the callback that sets this page as active in
    ↪   browsergym
    pass

# https://playwright.dev/python/docs/api/class-page#page-close
def tab_close():
    """
    Close the current tab.

    Examples:
        tab_close()
    """
    pass

#
↪   https://playwright.dev/python/docs/api/class-page#page-bring-to-front
def tab_focus(index: int):
    """
    Bring tab to front (activate tab).

    Examples:
        tab_focus(2)
```

```
        """
        pass

# https://playwright.dev/python/docs/input#upload-files
def upload_file(bid: str, file: str | list[str]):
    """
    Click an element and wait for a "filechooser" event, then
    ↪   select one
    or multiple input files for upload. Relative file paths are
    ↪   resolved
    relative to the current working directory. An empty list
    ↪   clears the
    selected files.

    Examples:
        upload_file("572", "my_receipt.pdf")
        upload_file("63", ["/home/bob/Documents/image.jpg",
        ↪   "/home/bob/Documents/file.zip"])
    """
    pass

# https://playwright.dev/python/docs/input#upload-files
def mouse_upload_file(x: float, y: float, file: str | list[str]):
    """
    Click a location and wait for a "filechooser" event, then
    ↪   select one
    or multiple input files for upload. Relative file paths are
    ↪   resolved
    relative to the current working directory. An empty list
    ↪   clears the
    selected files.

    Examples:
        mouse_upload_file(132.1, 547, "my_receipt.pdf")
        mouse_upload_file(328, 812,
        ↪   ["/home/bob/Documents/image.jpg",
        ↪   "/home/bob/Documents/file.zip"])
    """
    pass
```

## K    BROWSING AGENT DETAILS

The following shows an example prompt containing all the information required for the current step
to make a prediction about the next browsing actions. Note that we also instruct the agent to predict
multiple actions in one turn if the agent thinks they are meant to be executed sequentially without any
feedback from the page. This could save turns for common workflows that consist of a sequence of
actions on the same page without any observation change, such as filling the username and password
and submit in a login page.

```
# Instructions
Review the current state of the page and all other information to
↪   find the best possible next action to accomplish your goal.
↪   Your answer will be interpreted and executed by a program,
↪   make sure to follow the formatting instructions.
```

```
# Goal:
Browse localhost:8000, and tell me the ultimate answer to life. Do
↪  not ask me for confirmation at any point.

# Action Space

16 different types of actions are available.

noop(wait_ms: float = 1000)
    Examples:
        noop()

        noop(500)

send_msg_to_user(text: str)
    Examples:
        send_msg_to_user('Based on the results of my search, the
        ↪  city was built in 1751.')

scroll(delta_x: float, delta_y: float)
    Examples:
        scroll(0, 200)

        scroll(-50.2, -100.5)

fill(bid: str, value: str)
    Examples:
        fill('237', 'example value')

        fill('45', 'multi-line\nexample')

        fill('a12', 'example with "quotes"')

select_option(bid: str, options: str | list[str])
    Examples:
        select_option('48', 'blue')

        select_option('48', ['red', 'green', 'blue'])

click(bid: str, button: Literal['left', 'middle', 'right'] =
↪  'left', modifiers: list[typing.Literal['Alt', 'Control',
↪  'Meta', 'Shift']] = [])
    Examples:
        click('51')

        click('b22', button='right')

        click('48', button='middle', modifiers=['Shift'])

dblclick(bid: str, button: Literal['left', 'middle', 'right'] =
↪  'left', modifiers: list[typing.Literal['Alt', 'Control',
↪  'Meta', 'Shift']] = [])
    Examples:
        dblclick('12')

        dblclick('ca42', button='right')

        dblclick('178', button='middle', modifiers=['Shift'])
```

```
hover(bid: str)
    Examples:
        hover('b8')

press(bid: str, key_comb: str)
    Examples:
        press('88', 'Backspace')

        press('a26', 'Control+a')

        press('a61', 'Meta+Shift+t')

focus(bid: str)
    Examples:
        focus('b455')

clear(bid: str)
    Examples:
        clear('996')

drag_and_drop(from_bid: str, to_bid: str)
    Examples:
        drag_and_drop('56', '498')

upload_file(bid: str, file: str | list[str])
    Examples:
        upload_file('572', 'my_receipt.pdf')

        upload_file('63', ['/home/bob/Documents/image.jpg',
        ↪  '/home/bob/Documents/file.zip'])

go_back()
    Examples:
        go_back()

go_forward()
    Examples:
        go_forward()

goto(url: str)
    Examples:
        goto('http://www.example.com')

Multiple actions can be provided at once. Example:
fill('a12', 'example with "quotes"')
click('51')
click('48', button='middle', modifiers=['Shift'])
Multiple actions are meant to be executed sequentially without any
↪  feedback from the page.
Don't execute multiple actions at once if you need feedback from
↪  the page.

# Current Accessibility Tree:
RootWebArea 'The Ultimate Answer', focused
        [8] heading 'The Ultimate Answer'
        [9] paragraph ''
                StaticText 'Click the button to reveal the answer
                ↪  to life, the universe, and everything.'
        [10] button 'Click me', clickable
```

```
# Previous Actions
goto('http://localhost:8000')

Here is an example with chain of thought of a valid action when
↪  clicking on a button:
"
In order to accomplish my goal I need to click on the button with
↪  bid 12
```click("12")```
```

And an example response to the above prompt is:

```
In order to accomplish my goal, I need to click on the button with
↪  bid 10 to reveal the answer to life, the universe, and
↪  everything.
```click("10")```
```

For the evaluation on WebArena benchmark, since some of the tasks require checking for answer exact match on the agent's message back to the user, we add the following instruction to let the agent reply with only a concise answer string when messaging the user to prevent the agent from failing the test due to extra text:

```
Here is another example with chain of thought of a valid action
↪  when providing a concise answer to user:
"
In order to accomplish my goal I need to send the information
↪  asked back to the user. This page list the information of HP
↪  Inkjet Fax Machine, which is the product identified in the
↪  objective. Its price is $279.49. I will send a message back to
↪  user with the answer.
```send_msg_to_user("$279.49")```
"
```

