# OpenReview forum: "OpenHands: An Open Platform for AI Software Developers as Generalist Agents"
_ICLR.cc/2025/Conference — ICLR 2025 Poster_

### Official Review · Reviewer_nJqs · 2024-10-30

**Soundness:** 2
**Presentation:** 3
**Contribution:** 3
**Rating:** 6
**Confidence:** 5

**Summary:**

This paper describes OpenHands, fka OpenDevin, a platform to develop LLM agents which can take meaningful actions using computers, such as browsing, coding, and interacting with a command line.

Section 2 describes the architecture of the platform, detailing the state and event streams, actions and observations, and an extensible library of tools. Agent delegation is also mentioned.

Section 3 mentions some notable agents implemented using their platform.

Section 4 presents results of many evaluatory benchmarks, showing the OpenHands has a broad and general level of competence across domains including coding and browsing.

**Strengths:**

OpenHands is an excellent and significant LLM-agent platform, seeing use e.g. as one of the reference agents in OpenAI's MLE-bench https://arxiv.org/abs/2410.07095 , and there is value in having this scaffold's architecture explained and benchmark performance reported in a peer-reviewed form.


- General competence across three different domains (SWE, browsing, and 'misc') is significant

- Solve rate of 26% on SWE-bench is competitive, as are many of the results suggested by the tables (though, see below, I have questions around like-for-like comparison)

- The paper provides a great many benchmark results

- OpenHands has many useful features, such as the API server being inside the docker container, the ability to delegate subtasks to other agents, and an extensible library of tools

**Weaknesses:**

The paper is an announcement of a software platform, more than it is a description of a contribution to a research area. Normally I'd expect a paper to look like a research question, with some motivation, experimental design, and results. This paper shows little, if any, scientific exploration, instead the paper gives a high-level overview of the architecture with little motivation, and then uncritically showcases performance on a broad range of existing benchmarks. I'm unsure whether this kind of project-announcement is a fit for the conference.

Some smaller points:

- Why do the (e.g.) GPQA results not compare like-for-like LLMs? I would want to see whether a CodeActAgent powered by a particular LLM was better or worse than the bare LLM, but the table compares (Llama 3 70b chat, GPT 3.5 turbo 16k, GPT 4) with (gpt-4o-mini-2024-07-18 , gpt-4o-2024-05-13 , claude-3-5-sonnet), so it's unclear how much of the difference is due to the scaffold and what is due to the choice of LLM. This is also true for many results in Tables 4 onwards, perhaps an unfortunate downside of comparing to results from the literature, but this limitation precluding rigorous comparison should at least be acknowledged in the text.

- A large fraction of the paper comprises short summaries of many different benchmarks used to evaluate the codebase. Since these summaries are not original work, I question whether so much of the paper should be dedicated to them. (As a minor point, I'd expect this exposition to be in a setup / methodology / problem-setting section, rather than in the results.) Instead, the authors could focus more on the creative decisions made during the creation of their platform.

**Questions:**

L305 - I'm uncertain what is meant by "we compare OpenHands to open-source reproducible baselines that do not perform manual prompt engineering specifically based on the benchmark content"

It seems surprising that OH CodeActAgent w. GPT-3.5 would get 2% on ToolQA, while ReAct (perhaps the most minimal form of agent scaffolding) would get 36%. Do you have any idea what's behind this performance difference?

Also surprising is that CodeActAgent delegating to BrowsingAgent does very slightly worse than straightforwardly using BA directly. Could you discuss this?

---

> ### Author Response · Authors · 2024-11-21
>
> We appreciate the reviewer’s thoughtful feedback and are pleased that the reviewer find our work well-motivated, well-written, and useful to the community.
>
> > The paper is an announcement of a software platform, more than it is a description of a contribution to a research area. Normally I'd expect a paper to look like a research question, with some motivation, experimental design, and results…. I'm unsure whether this kind of project-announcement is a fit for the conference….  and then uncritically showcases performance on a broad range of existing benchmarks.
>
> According to ICLR 2025 call for paper (https://iclr.cc/Conferences/2025/CallForPapers), OpenHands primarily falls under the category of “infrastructure, software libraries, hardware, etc.” where we consider the primary contribution be the infrastructure that supports development of generalist software (multi-)agent systems with runtime environment, interface, and evaluation.
> The demonstrated performance across a broad range of benchmarks covering different domains (e.g., software, browsing, general assistance) shows OpenHands’ ability to solve a wide range of tasks without changing the infra as well as the agent, and supports the potential of OpenHands as an infrastructure to support development of generalist digital agents.
>
> > Why do the (e.g.) GPQA results not compare like-for-like LLMs? I would want to see whether a CodeActAgent powered by a particular LLM was better or worse than the bare LLM, but the table compares (Llama 3 70b chat, GPT 3.5 turbo 16k, GPT 4) with (gpt-4o-mini-2024-07-18 , gpt-4o-2024-05-13 , claude-3-5-sonnet), so it's unclear how much of the difference is due to the scaffold and what is due to the choice of LLM. This is also true for many results in Tables 4 onwards, perhaps an unfortunate downside of comparing to results from the literature, but this limitation precluding rigorous comparison should at least be acknowledged in the text.
>
> Thanks for pointing this out, we acknowledge this limitation in our current evaluation and will update our next revision accordingly. We are actively working on expanding results for our evaluation benchmarks to include more LLM backbones. Nevertheless, we believe that our existing result does serve the purpose of demonstrating the generalist capabilities of OpenHands: it is able to solve a wide range of tasks, that used to require custom prompting, with the SAME agent scaffold and system message and achieve non-trivial performance compared to prior work that applies custom prompting.
>
> > A large fraction of the paper comprises short summaries of many different benchmarks used to evaluate the codebase. Since these summaries are not original work, I question whether so much of the paper should be dedicated to them…
>
> We appreciate the reviewer’s feedback and will be sure to include a more extensive discussion on the results in the next revision. We spend a lot of effort trying to cut down the summaries for original work, however, we re-adapt some settings about these existing benchmarks to bring them into OpenHands evaluation harness and we feel it is still necessary to include these in the paper.
>
>
> > L305 - I'm uncertain what is meant by "we compare OpenHands to open-source reproducible baselines that do not perform manual prompt engineering specifically based on the benchmark content"
>
> We consider the following criteria for including baseline for browsing benchmarks:
> - The baseline needs to have open source reproducible code so we can verify its result. There were some WebArena numbers publicized only on Twitter, without actual open-source code to reproduce the result.
> - We also disqualified baselines that perform manual prompt engineering on the testset of a benchmark (e.g., SteP uses handcrafted  "routines" or "workflow" library specifically designed for WebArena websites)

---

> ### Author Response · Authors · 2024-11-21
>
> > It seems surprising that OH CodeActAgent w. GPT-3.5 would get 2% on ToolQA, while ReAct (perhaps the most minimal form of agent scaffolding) would get 36%. Do you have any idea what's behind this performance difference?
>
> We are using the exact SAME system prompt to evaluate all the tasks for CodeActAgent (the generalist) to make sure it general enough to be able to handle a wide range of tasks. This includes distractors to weaker models (e.g., instruction about how to use browser is typically not relevant to solve a ToolQA problem), hence we are seeing a lower performance on weaker models like GPT-3.5 where we observe they would be overwhelmed by the system message and get stuck into loops. On the contrary, stronger models like gpt-4o demonstrate reasonable performance.
>
>
> > Also surprising is that CodeActAgent delegating to BrowsingAgent does very slightly worse than straightforwardly using BA directly. Could you discuss this?
>
> We measure performance of the browsing delegation by asking CodeActAgent to produce the correct agent delegation action and evaluate the correctness of the delegation request. In some cases, CodeActAgent would not be able to pass the exact browsing task to BrowserAgent, which results in the minor performance degradation.

---

> ### Comment · Reviewer_nJqs · 2024-11-22
>
> I welcome the pointer to the Call for Papers, and specifically the pointer to the category of “infrastructure, software libraries, hardware, etc.” I agree that OpenHands falls into this category, and so my main concern is allayed.
>
> The clarification around L305 is helpful, thank you. Perhaps this is worth including in the next revision.
>
> The 2% -> 36% is surprising enough that the explanation you give about irrelevancies in long prompts distracting weaker models (very clear now, thank you) warrants inclusion in the next version, as perhaps does the points about CAA sometimes passing an inexact task to BA.
>
> For the other points, I welcome the next revision.

---

### Official Review · Reviewer_WpcF · 2024-11-01

**Soundness:** 3
**Presentation:** 4
**Contribution:** 4
**Rating:** 8
**Confidence:** 4

**Summary:**

OpenHands is an open-source platform for the development of general AI agents. This paper details the motivation behind OpenHands, the architecture of the platform, how a user might implement an agent using OpenHands, description of some existing baseline agents, and comparison of some of the OpenHands agents against other open-source agents on software, web, and other assisting tasks. Selected results show that their agents perform reasonably well across a wide range of these tasks (generally underperform the specialized agents).

**Strengths:**

OpenHands is a valuable contribution to the research community. The OpenAgents platform will surely be used to develop many future agents, and be referred to as a baseline for other agent-developing research projects. Tables 3, 4, and 5 present thorough experimentation of their agent(s) compared to many other baselines on various tasks.

**Weaknesses:**

While the value of OpenHands is apparent, the research question of this paper is unclear. If it is that they can develop a general agent that performs better across a range of tasks than any existing agent does, it is my judgement that the results and analysis do not explore this sufficiently to warrant a research contribution. To do so, discussion and ablations should be included for the design decisions of OH Browsing Agent v*, OH CodeAgent v*, OH GPTSwarm v*, etc. To then isolate the effect of those decisions, the choice of backend model should be held consistent to those of the baselines against which the OH agents are compared. Error analysis of results comparing OH to baselines would be another welcome contribution that has been omitted from this paper.

**Questions:**

Can we further motivate why we want a generalist agent? If we have specialized agents for different tasks, can we just have a coordinator deploy specialized agents where they perform best?

---

> ### Author Response · Authors · 2024-11-21
>
> We appreciate the reviewer’s thoughtful feedback and are pleased that the reviewer finds our work sound, well-written, and valuable to the community.
>
> > While the value of OpenHands is apparent, the research question of this paper is unclear. If it is that they can develop a general agent that performs better across a range of tasks than any existing agent does, it is my judgement that the results and analysis do not explore this sufficiently to warrant a research contribution. To do so, discussion and ablations should be included for the design decisions of OH Browsing Agent v*, OH CodeAgent v*, OH GPTSwarm v*, etc. To then isolate the effect of those decisions, the choice of backend model should be held consistent to those of the baselines against which the OH agents are compared. Error analysis of results comparing OH to baselines would be another welcome contribution that has been omitted from this paper.
>
> Based on the ICLR 2025 call for papers (https://iclr.cc/Conferences/2025/CallForPapers), we consider OpenHands primarily falls under the category of “infrastructure, software libraries, hardware, etc,” where we consider the primary contribution be the **infrastructure that supports development of generalist software (multi-)agent systems** with runtime environment, interface, and evaluation. While analyzing agent design decisions would be valuable, we have limited space in the paper and we barely fit the description of all existing OpenHands components and evaluations with current page limits. Therefore, we would refer to the individual agent papers for detailed analysis (e.g., CodeAct, GPTSwarm) and focus on infrastructure that enables such agentic research and evaluation in this paper.
>
> > Can we further motivate why we want a generalist agent? If we have specialized agents for different tasks, can we just have a coordinator deploy specialized agents where they perform best?
>
> While having a “specialized agent” + multi-agent system could be good for individual task performance (and maybe cost, since specialized agent could use specialized model which can be made small and efficient), they are not without practical drawbacks:
>
> - **Real-world tasks are not always perfectly decomposable into multi-agent + specialized agent setting**: For example, for coding tasks, a multi-agent system CodeR [1] has five specialized agents (Manager, Reproducer, Fault Localizer, Editor, Verifier). For instance, what if the verifier, which is responsible of outputting yes/no decision of whether the issue is resolved or not, wants to perform file localization to be sure that it has verified the answer properly? Because these agents are completely separate, the verifier would not have access to the tools necessary to do its job.
>
> - **Challenging to preserve context in communication**: Multi-agent systems typically pass information between multiple agents, but this can be a source of information loss. Following the above CodeR example, if the fault localizer passes only a summary of its work on to the further agents, it often results in the loss of important contextual information that could be useful to the downstream agents for their jobs.
>
> - **Maintainability**: Finally, each of these specialized agents typically has its own separate code base, or at least a separate prompt. Because of this, multi-agent systems can have larger and more complex codebases. And the codebase could be constantly growing as we require more capability from the agent – making maintenance hard and eventually causing reliability issues with the codebase.
>
> Due to the above potential drawbacks, we are motivated to make our infrastructure general enough to support developing and researching both generalist and specialist agents so we can further explore both the advantages and disadvantages of these directions going forward.
>
> [1] CodeR: Issue Resolving with Multi-Agent and Task Graphs

---

> > ### Comment · Reviewer_WpcF · 2024-11-22
> >
> > Thank you for the response. My concerns about whether ICLR is an appropriate venue for OpenHands has been alleviated. I still believe analysis would strengthen the contribution of this work as a research paper, but I recognize that space limitations force a tradeoff between volume of evaluated agents and detail of analysis. I have raised my score accordingly.

---

### Official Review · Reviewer_4sob · 2024-11-04

**Soundness:** 3
**Presentation:** 3
**Contribution:** 4
**Rating:** 8
**Confidence:** 4

**Summary:**

The paper presents OpenHands, an open-source platform designed to facilitate the development of AI agents that interact with the world through software interfaces such as code execution, command-line operations, and web browsing. A key feature of OpenHands is AgentHub, a platform where users can contribute their own agents (architecture) to the community, fostering collaborative development of agent architectures. The platform provides a simple agent abstraction, making it accessible for users to create and extend agents, and provides a modular architecture for agent interactions and task execution, including an event-driven state management system, multi-agent coordination, and an extensible agent skills library. OpenHands aims to enable agents to even create tools by themselves. The authors describe the architecture, implementation details, and evaluate the platform across multiple benchmarks, including software engineering tasks and web browsing scenarios. It's kind of a combination of AutoGen and BrowserGym.

**Strengths:**

Originality:
OpenHands uniquely consolidates multiple agent capabilities: coding, command-line interaction, web browsing, and multi-agent collaboration, within a single, open-source platform. The introduction of AgentHub allows users to contribute their own agents, promoting community collaboration and expanding the platform's versatility. This integration distinguishes OpenHands from existing frameworks, although IMO some of the main ones are absent from the comparison (see questions).

Quality:
The methodological approach is solid, featuring a well-defined architecture that includes an event-driven state management system and a secure, sandboxed runtime environment. The use of an extensible agent skills library enhances flexibility, allowing agents to perform complex tasks and even create tools themselves. The agent abstraction is designed to be simple, enabling users to easily create and extend agents.

Clarity:
The paper is generally well-written and logically structured, making it accessible to both practitioners and researchers. Key components of the platform, such as the agent abstraction, runtime environment, and evaluation framework, are explained clearly. Figures and tables effectively illustrate concepts and results.

Significance:
By providing an open-source, MIT-licensed platform with contributions from a large community, OpenHands has the potential to significantly impact the development and evaluation of AI agents. It introduces a varied range of datasets for each general agent domain, although it could be expanded in the browser domain (see weaknesses). Evaluating multiple close-source LLMs on each task in addition to the average costs of evaluating benchmarks is a valuable addition that offers practical insights.

**Weaknesses:**

Limited Novelty in Certain Aspects: While OpenHands integrates various functionalities, much of the work appears to assemble existing components from the domain into one place. Some features, like code execution in a sandbox and web browsing agents, are present in other platforms. The paper could better articulate how OpenHands distinguishes itself from similar frameworks like LangChain, DSPy, or AutoGen.

Basic Implemented Agents: The currently implemented agents are relatively basic. For example, the Browsing Agent is based on WebArena's agent, which is very simple to implement and not competitive with current SOTA agents. There are more advanced multi-agent architectures available that could be implemented in AgentHub to better demonstrate the system's capabilities in facilitating the general agents' development.

Outdated Datasets: The evaluation includes MiniWoB++, which is an outdated dataset. There are newer datasets, such as WorkArena, WorkArena++, and ST-WebAgentBench, that are more relevant. The last test agents in more complex, restricted environments using policy hierarchies. Incorporating these in the framework would strengthen the evaluation.

Evaluation Depth: The evaluation, while broad, could benefit from deeper analysis. Incorporating an automated data collector to track agent success and failure would enhance insight into agent performance and make the framework more complete.

**Questions:**

Distinction from Existing Platforms: How does OpenHands fundamentally differ from other agent development platforms like LangGraph, CrewAI and DSPy for multi-agent collaboration or BrowserGym for benchmark evaluation? Could the authors elaborate on the unique features or advantages that OpenHands offers over these frameworks? Including these comparisons in Table 1 would provide a more comprehensive overview.

Agent Performance Analysis: Can the authors provide a more detailed explanation about the data collector in the framework? Specifically, what makes the framework the best solution to develop an agent when one approaches such a task?
Another question is: how will the platform address this in future iterations?

Handling of Observations: How does the platform treat observations and their full composition? Can the authors clarify how observations are managed and utilized by agents? Providing more details on the configuration and usage of the event stream would enhance understanding, especially for developers less familiar with event-driven systems.

Security Considerations: Given that the platform allows the execution of arbitrary code in a sandboxed environment, what security measures are in place to prevent potential exploits or breaches? Has the sandboxing been tested against known vulnerabilities? Discussing security protocols would demonstrate the platform's reliability.

Web Page Understanding Techniques: Regarding the web experiments, it is not clear which page understanding techniques are used to interpret interactable elements and web pages. Does the framework provide screen/page understanding capabilities, or should a new user implement their own methods? Clarifying this would help users understand the platform's capabilities in web interaction tasks.

---

> ### Author Response · Authors · 2024-11-21
>
> We appreciate the reviewer’s detailed feedback and are pleased that the reviewer finds our work solid, well-written, and useful to the community.
>
> > …Some features, like code execution in a sandbox and web browsing agents, are present in other platforms. The paper could better articulate how OpenHands distinguishes itself from similar frameworks like LangChain, DSPy, or AutoGen… Distinction from Existing Platforms: How does OpenHands fundamentally differ from other agent development platforms like LangGraph, CrewAI and DSPy for multi-agent collaboration or BrowserGym for benchmark evaluation? ….. Specifically, what makes the framework the best solution to develop an agent when one approaches such a task?..... Distinction from Existing Platforms: How does OpenHands fundamentally differ from other agent development platforms like LangGraph, CrewAI and DSPy for multi-agent collaboration or BrowserGym for benchmark evaluation?
>
> We describe the differences between OpenHands and some popular existing agent frameworks in Table 1. We did not include DSPy as it is focusing on end-to-end prompt optimization and less relevant as an “agent framework”. We will include CrewAI and BrowserGym in the next revision. Here’s a high-level overview of the differences between OpenHands and CrewAI & BrowserGym
>
> - CrewAI largely focuses on orchestrating multi-agent communications, but lacks features on the sandboxed terminal, browser, and file system compared to OpenHands - which we will explain more in the following paragraph. It also doesn’t include a comprehensive evaluation harness to support development.
> - BrowserGym, on the other hand, is a platform specifically designed for browsing agents - it lacks the feature of sandboxed code execution and file system, and contains benchmarks only for web browsing, whereas OpenHands aims to build a general software engineering agent that’s capable of doing much more than browsing.
>
> That is, existing frameworks mostly cover a subset of OpenHands features, whereas OpenHands smoothly integrates the entire workflow of AI agent development and deployment:
>
> **(development & deployment) Sandboxed Runtime Environment that can build on arbitrary docker images**: Different from most agent frameworks that don’t provide this feature, OpenHands provides a complete integrated runtime environment, which is non-trivial to implement and scale. Compared to existing frameworks:
>
> - The OpenHands runtime environment automatically builds supports for the terminal, file system and editor and browser automatically on top of arbitrary docker images that the user provides - this allows OpenHands to run on any user-provided repository-specific development image, which other existing frameworks don’t support to the best of our knowledge.
> - LangChain/LangGraph has only very limited support for runtime (e.g., only supports code execution - but not bash-like terminal commands), and most of this support requires using third-party non-open-source solutions (https://python.langchain.com/docs/integrations/tools/#code-interpreter).
> - AutoGen does have limited support for python, bash execution. But different from the OpenHands implementation, to the best of our knowledge, AutoGen’s [python & bash execution](https://github.com/microsoft/autogen/blob/b65269b8f81296d94f67011a49d8012e718ea749/python/packages/autogen-ext/src/autogen_ext/code_executors/_docker_code_executor.py#L217) [based on docker exec](https://docker-py.readthedocs.io/en/stable/containers.html#docker.models.containers.Container.exec_run) is not stateful: If the agent changed to a directory from a previous action (e.g., cd aaa/), the next bash action will NOT be executed in the directory aaa/ - this is non-trivial to implement and can be a major confusion point for an agent if not implemented. To the best of our knowledge, OpenHands is the only one comprehensive open-source agent framework that supports stateful bash command execution via pexcept.
> - CrewAI has limited support for sandboxed code execution (https://github.com/crewAIInc/crewAI-tools/blob/afa7e6a64347e09003088c400afcf4503433f2a2/crewai_tools/tools/code_interpreter_tool/code_interpreter_tool.py#L107), but it is restricted to stateless Python code, whereas OpenHands support both stateful bash & python (jupyter) execution within sandboxed runtime.
> - AutoGen and CrewAI also have limited support for browsing (https://microsoft.github.io/autogen/0.2/docs/reference/browser_utils/abstract_markdown_browser, https://github.com/crewAIInc/crewAI-tools/blob/afa7e6a64347e09003088c400afcf4503433f2a2/crewai_tools/tools/browserbase_load_tool/browserbase_load_tool.py#L47-L50), however, they only support one feature which is “getting the content of a webpage in a markdown format.” On the other hand, OpenHands sits on top of BrowserGym, not only supports getting markdown format, also allows the agent to interact with the web page directly by observing screenshots, clicking buttons, inputting text, etc.

---

> > ### Author Response · Authors · 2024-11-21
> >
> > More importantly, OpenHands integrates all these important action spaces (terminal, browser, IPython, file editor) into a shared file system inside the docker container; yet existing frameworks (including LangChain, AutoGen, CrewAI) typically implement these sandboxed features in a fragmented way: the agent can individually access to these action spaces, but cannot easily share results directly between components. For example, the sandboxed bash terminal can execute agent’s command, BUT it cannot interact with user’s files; the browser can view web pages from internet, but cannot access things in a local file system. That is, existing frameworks typically sandbox the code execution, but not the file system. This greatly restricts the flexibility of an agent: for example, it would be challenging for an agent to mount a repository on a user’s environment, modify it, install dependencies for it, and execute some of the code there. On the other hand, OpenHands was designed as a software engineer agent that integrates all these capabilities within ONE docker sandbox, which allows seamless collaboration between all these tools.
> >
> > **(development) Standardized Evaluation**: Different from most existing frameworks, OpenHands, with the goal of developing AI software engineer as a generalist, has an evaluation harness containing 15 challenging benchmarks covering different capabilities – and this list of evaluations keep expanding as OpenHands continues to evolve as an open-source project. Since the paper submissions, more challenging benchmarks are being added to the OpenHands ecosystem (e.g., ScienceAgentBench [1], MLE-Bench [2], AiderBench [3], Commit0 [4]). This suite of comprehensive evaluations helps measure and inform the development of OpenHands. Out of all other frameworks we compared in Table 1, only AutoGen has an evaluation harness with only 4 evaluations with relatively limited coverage of domains.
> >
> > [1] ScienceAgentBench: Toward Rigorous Assessment of Language Agents for Data-Driven Scientific Discovery
> >
> > [2] MLE-Bench: Evaluating Machine Learning Agents on Machine Learning Engineering
> >
> > [3] https://aider.chat/docs/benchmarks.html
> >
> > [4] https://github.com/commit-0/commit0
> >
> >
> > **(deployment) User interface**: Different from all the frameworks we compared with, as shown in Figure 1, OpenHands has an immediately useable user interface for all the runtime action spaces we’ve introduced. It allows the user to view files, check executed bash commands/Python code, observe the agent’s browser activity, and directly interact with the agent via conversation.
> >
> > We thank the reviewer for pointing out this confusion, we will include the above discussion in the next revision to make it clearer.
> >
> > > Basic Implemented Agents: … There are more advanced multi-agent architectures available that could be implemented in AgentHub to better demonstrate the system's capabilities… the future. … Evaluation Depth: The evaluation, while broad, could benefit from deeper analysis. Incorporating an automated data collector to track agent success and failure would enhance insight into agent performance and make the framework more complete.
> >
> > While it is definitely possible to implement complex multi-agent systems in OpenHands and perform evaluation result analysis, limited by max page limit, it was not the main scope of this paper.  Based on the ICLR 2025 call for papers (https://iclr.cc/Conferences/2025/CallForPapers), we consider OpenHands primarily falling under the category of “infrastructure, software libraries, hardware, etc,” where we consider the primary contribution to be the *infrastructure that supports the development of (multi-)agent systems* with runtime environment, interface, and evaluation.
> >
> > > Outdated Datasets: The evaluation includes MiniWoB++, which is an outdated dataset. There are newer datasets, such as WorkArena, WorkArena++, and ST-WebAgentBench, that are more relevant. The last test agents in more complex, restricted environments using policy hierarchies. Incorporating these in the framework would strengthen the evaluation.
> >
> > As described earlier, since the submission of the paper, more challenging benchmarks are being added to the OpenHands ecosystem  (e.g., ScienceAgentBench [1], MLE-Bench [2], AiderBench [3], Commit0 [4]). Being deeply integrated with browsergym, going forward, it will be relatively easy to include newer benchmarks like WorkArena into the evaluation harness, which we do plan to do in the future as the framework continues to evolve.
> >
> > [1] ScienceAgentBench: Toward Rigorous Assessment of Language Agents for Data-Driven Scientific Discovery
> >
> > [2] MLE-Bench: Evaluating Machine Learning Agents on Machine Learning Engineering
> >
> > [3] https://aider.chat/docs/benchmarks.html
> >
> > [4] https://github.com/commit-0/commit0

---

> > > ### Author Response · Authors · 2024-11-21
> > >
> > > > Handling of Observations: How does the platform treat observations and their full composition? Can the authors clarify how observations are managed and utilized by agents? Providing more details on the configuration and usage of the event stream would enhance understanding, especially for developers less familiar with event-driven systems.
> > >
> > > For browsing, the observations contain multiple aspects of the current browser state, including the open tabs’ URLs, the raw HTML source of the current webpage, the screenshot of the current viewport, the processed accessibility tree of the current webpage, and the raw text of the current webpage. These observations are all sent as the observation events and it’s up to the agent implementation on how and which of these information to actually use to predict the next steps. Similarly, the observation from the terminal and IPython shell are directly sent to the agent as raw string, with the exception that the terminal included more metadata (e.g., error code) in the output.
> > >
> > > > Web Page Understanding Techniques: Regarding the web experiments, it is not clear which page understanding techniques are used to interpret interactable elements and web pages. Does the framework provide screen/page understanding capabilities, or should a new user implement their own methods? Clarifying this would help users understand the platform's capabilities in web interaction tasks.
> > >
> > > We incorporated BrowserGym (https://github.com/ServiceNow/BrowserGym)’s handling of web page processing, and interactable elements and web page textualization, accessibility tree generation, as well as a better format suited for LLM consumption is included right out of the box for the user. The user can choose to use our provided processing techniques, but also, since we provide all the raw observations of the web page, the users implementing their own agent are free to use whatever processing they see fit. That’s also part of the advantage of our framework, it’s batteries included, but also fully customizable.
> > >
> > > > Security Considerations: Given that the platform allows the execution of arbitrary code in a sandboxed environment, what security measures are in place to prevent potential exploits or breaches? Has the sandboxing been tested against known vulnerabilities? Discussing security protocols would demonstrate the platform's reliability.
> > >
> > > As discussed in detail in Section 2.2, we use docker sandbox to containerized runtime which is responsible for all code execution. Docker sandboxes have been widely used in the industry and have been thoroughly tested and fixed against different vulnerabilities – we refer to docker’s security announcements for more information: https://docs.docker.com/security/security-announcements/
> > >
> > > > Can the authors provide a more detailed explanation about the data collector in the framework? … Another question is: how will the platform address this in future iterations?
> > >
> > > We’d appreciate it if the reviewer could clarify more about “the data collector” – it is unclear to us what the data collector refers to. We would appreciate it if the reviewer could point to the specific section number for the data collector and we would be happy to answer questions about it.

---

> > > > ### Comment · Reviewer_4sob · 2024-11-26
> > > > **Official Comment By Reviewer**
> > > >
> > > > Thank you for your detailed and comprehensive responses to my review. I appreciate the time and effort you've taken to address my questions and concerns.
> > > >
> > > > Your clarifications on how OpenHands distinguishes itself from existing platforms like LangChain, CrewAI, DSPy, and BrowserGym have been particularly helpful. The integrated sandboxed runtime environment, with support for stateful bash command execution and seamless collaboration between tools within a shared file system, sets your platform apart. I also recognize the significance of your standardized evaluation harness, which includes a broad suite of challenging benchmarks and continues to expand.
> > > >
> > > > I'm pleased to hear about your plans to incorporate newer datasets and benchmarks, such as WorkArena and others, into the framework. Your explanations regarding the handling of observations, web page understanding techniques, and security considerations have addressed my concerns effectively.
> > > >
> > > > Overall, your responses have alleviated the issues I raised, and I believe that OpenHands makes a valuable contribution to the development and evaluation of AI agents. I support the acceptance of your paper and look forward to seeing the impact your platform will have on the community.

---

### Official Review · Reviewer_ujp2 · 2024-11-07

**Soundness:** 3
**Presentation:** 3
**Contribution:** 3
**Rating:** 6
**Confidence:** 4

**Summary:**

This paper introduces OpenHands, is a community-driven platform designed to advance the development of AI agents that interact with the world in terms of code writing, command-line interaction, and web browsing. OpenHands includes an evaluation framework for benchmarking agent performance on tasks like software engineering and web browsing. It supports various LLMs and sandboxed environments for running these AI agents and conducts extensive evaluation of different agents in various benchmarks.

**Strengths:**

- The motivation to build an open-sourced agent platform is good.
- The agent interface, with capabilities for code execution and web browsing, is impressive.
- Evaluations conducted across various agents and benchmarks are thorough and extensive.

**Weaknesses:**

N/A

**Questions:**

This paper introduces an open platform with the event-driven architecture to advance the AI agent development, especially focusing on coding and web browsing. It also conducts extensive evaluations of heterogeneous agents on various benchmarks. This is a solid contribution to the LLM agent community. I have some questions about the implementation details.
- The authors mention that event streaming enables human intervention in agent tasks, which I believe is a crucial step for calibrating agent outputs. I wonder whether human intervention is enabled during the evaluation of agents and is there a mechanism to determine whether human intervention is required?
- Regarding the multi-agent delegation feature, I’m unclear on how agents collaborate. Is there an orchestrator agent responsible for task delegation, or is this managed through event streaming? If it’s the latter, how are tasks assigned and dispatched, and where do agents deliver their outputs?

---

> ### Author Response · Authors · 2024-11-21
>
> We appreciate the reviewer’s thoughtful feedback and are pleased that the reviewer found our work well-motivated, well-written, and useful to the community.
>
> > The authors mention that event streaming enables human intervention in agent tasks, which I believe is a crucial step for calibrating agent outputs. I wonder whether human intervention is enabled during the evaluation of agents and is there a mechanism to determine whether human intervention is required?
>
> During evaluation, following the existing benchmark setting, human intervention is disabled to ensure consistent and reproducible results. For example, the original SWE-Bench setup does not consider human inputs during evaluation, hence we prompt the agent to not interact with humans while solving a given SWE-Bench task. When it did try to interact with the user, we provided pre-determined feedback “Please continue working on the task on whatever approach you think is suitable. If you think you have solved the task, please first send your answer to user through message and then finish the interaction.” – this prompts the agent to either continue to solve the problem without human intervention, or finish the interaction.
>
> Regarding a mechanism to determine whether human intervention is required, typically the agent (and the underlying foundation model) is responsible for making such decisions (i.e., it only generates text without action for the next step). Based on our observation, it typically happens when the agent finishes the task, or gets stuck into some problems it cannot resolve.
>
> > Regarding the multi-agent delegation feature, I'm unclear on how agents collaborate. Is there an orchestrator agent responsible for task delegation, or is this managed through event streaming? If it's the latter, how are tasks assigned and dispatched, and where do agents deliver their outputs?
>
> In OpenHands, agent collaboration is implemented through an event streaming mechanism rather than a single orchestrator agent. When an agent needs to delegate a task, it submits an AgentDelegateAction to the event stream. A central controller then dispatches this request to the appropriate child agent and manages the return of results through the same event stream.
> Child agents are typically specialized (e.g., for browsing) and receive only their specific tasks and necessary context. This design ensures a clear separation of responsibilities while enabling flexible collaboration patterns. For example, our DelegatorAgent in our agenthub is an orchestrator agent that coordinates between three specialized agents for repository analysis, coding, and verification, while our ManagerAgent maintains a catalog of micro-agents defined in markdown and delegates tasks based on their capabilities. The ManagerAgent analyzes each micro-agent's description and constraints to select the most suitable one for a given task or rejects the task if no micro-agent is capable of handling it.

---

### Meta-Review · Area_Chair_5zT5 · 2024-12-16

**Metareview:**

After reading the reviewers' comments and reviewing the paper, we recommend acceptance - Oral presentation.

Below a more detailed description of the paper.

The paper introduces OpenHands, a community-driven platform designed to advance the development of AI agents that interact with the world in terms of code writing, command-line interaction, and web browsing, and it constitutes a good contribution to “infrastructure, software libraries, hardware, etc.”

The key strengths (S#) of the paper are as follows:

- (S1)	The OpenHands platform may be used to develop many future agents, and be referred to as a baseline for other agent-developing research projects. Tables 3, 4, and 5 present thorough experimentation of their agent(s) compared to many other baselines on various tasks.
- (S2)	OpenHands is original: it uniquely consolidates multiple agent capabilities such as coding, command-line interaction, web browsing, and multi-agent collaboration, within a single, open-source platform. The introduction of AgentHub allows users to contribute their own agents, promoting community collaboration and expanding the platform's versatility. This integration distinguishes OpenHands from existing frameworks, although IMO some of the main ones are absent from the comparison
- (S3)	The methodological approach is solid, featuring a well-defined architecture that includes an event-driven state management system and a secure, sandboxed runtime environment. The use of an extensible agent skills library enhances flexibility, allowing agents to perform complex tasks and even create tools themselves. The agent abstraction is designed to be simple, enabling users to easily create and extend agents.
- (S4)	By providing an open-source, MIT-licensed platform with contributions from a large community, OpenHands has the potential to significantly impact the development and evaluation of AI agents.
(S5)	General competence across three different domains (SWE, browsing, and 'misc') is significant.
(S6)	OpenHands has many useful features, such as the API server being inside the docker container, the ability to delegate subtasks to other agents, and an extensible library of tools.

Some of the key weaknesses (W#) are as follows:

- (W1)	Limited Novelty in Certain Aspects: While OpenHands integrates various functionalities, much of the work appears to assemble existing components from the domain into one place. Some features, like code execution in a sandbox and web browsing agents, are present in other platforms. The paper could better articulate how OpenHands distinguishes itself from similar frameworks like LangChain, DSPy, or AutoGen.
- (W2)	Basic Implemented Agents: The currently implemented agents are relatively basic. For example, the Browsing Agent is based on WebArena's agent, which is very simple to implement and not competitive with current SOTA agents. There are more advanced multi-agent architectures available that could be implemented in AgentHub to better demonstrate the system's capabilities in facilitating the general agents' development.
- (W3)	Outdated Datasets: The evaluation includes MiniWoB++, which is an outdated dataset. There are newer datasets, such as WorkArena, WorkArena++, and ST-WebAgentBench, that are more relevant. The last test agents in more complex, restricted environments using policy hierarchies. Incorporating these in the framework would strengthen the evaluation.
- (W4)	Evaluation Depth: The evaluation, while broad, could benefit from deeper analysis. Incorporating an automated data collector to track agent success and failure would enhance insight into agent performance and make the framework more complete.

**Additional Comments On Reviewer Discussion:**

The authors have been proactive in addressing the comments raised by the reviewers, and the reviewers were well engaged responding to the authors.

We agree with the reviewers comments, and recommendations, noting some of the weaknesses that we believe may remain and are mentioned in the metareview.

No ethics review raised by the reviewers, and we agree with them.

---

### Decision · Program_Chairs · 2025-01-22

Accept (Poster)